# Characterizing the Randot Preschool stereotest: Testability, norms, reliability, specificity and sensitivity in children aged 2-11 years

**Jenny C. A. Read**[1]*, **Sheima Rafiq**[1], **Jess Hugill**[1], **Therese Casanova**[1], **Carla Black**[1], **Adam O'Neill**[1], **Vicente Puyat**[1], **Helen Haggerty**[2], **Kathryn Smart**[2], **Christine Powell**[2], **Kate Taylor**[2], **Michael P. Clarke**[1,2], **Kathleen Vancleef**[1]

1 Institute of Neuroscience, Newcastle University, Newcastle upon Tyne, England, United Kingdom,
2 Newcastle Eye Centre, Royal Victoria Infirmary, Newcastle upon Tyne Hospitals NHS Trust, Newcastle upon Tyne, England, United Kingdom

* jenny.read@ncl.ac.uk

**Data Availability Statement:** All data, along with R markdown code to analyse them and produce the figures in the paper, are available at the following link: https://doi.org/10.25405/data.ncl.9755045.

## Abstract

### Purpose

To comprehensively assess the Randot Preschool stereo test in young children, including testability, normative values, test/retest reliability and sensitivity and specificity for detecting binocular vision disorders.

### Methods

We tested 1005 children aged 2–11 years with the Randot Preschool stereo test, plus a cover/uncover test to detect heterotropia. Monocular visual acuity was assessed in both eyes using Keeler Crowded LogMAR visual acuity test for children aged 4 and over.

### Results

**Testability** was very high: 65% in two-year-olds, 92% in three-year-olds and ~100% in older children. **Normative values**: In 389 children aged 2–5 with apparently normal vision, 6% of children scored nil (stereoblind). In those who obtained a threshold, the mean log threshold was 2.06 $\log_{10}$ arcsec, corresponding to 114 arcsec, and the median threshold was 100 arcsec. Most older children score 40 arcsec, the best available score. We found a small sex difference, with girls scoring slightly but significantly better. **Test/retest reliability:** ~99% for obtaining any score vs nil. Agreement between stereo thresholds is poor in children aged 2–5; 95% limit of agreement = 0.7 $\log_{10}$ arcsec: five-fold change in stereo threshold may occur without any change in vision. In children over 5, the test essentially acts only as a binary classifier since almost all non-stereoblind children score 40 arcsec. **Specificity** (true negative rate): >95%. **Sensitivity** (true positive rate): poor, <50%, i.e. around half of children with a demonstrable binocular vision abnormality score well on the Randot Preschool.

**Funding:** This manuscript presents independent research commissioned by the Health Innovation Challenge Fund (HICF-R8-442 and WT102565/z/13/z to JCAR and MPC), a parallel funding partnership between the Wellcome Trust (wellcome.ac.uk) and the UK Department of Health (www.gov.uk/government/organisations/department-of-health-and-social-care). The views expressed in this manuscript are those of the authors and not necessarily those of the Wellcome Trust or the Department of Health. The funders played no role in the study design, data collection and analysis, decision to publish, or preparation of the manuscript.

**Competing interests:** The authors have developed their own stereotest, which has recently been licensed to a company. The authors have no personal financial interest in this or any relevant product. This does not alter our adherence to PLOS ONE policies on sharing data and materials.

## Conclusions

The Randot Preschool is extremely accessible for even very young children, and is very reliable at classifying children into those who have any stereo vision vs those who are stereoblind. However, its ability to quantify stereo vision is limited by poor repeatability in children aged 5 and under, and a very limited range of scores relevant to children aged over 5.

## Introduction

Stereotests assess binocular visual function by measuring the smallest depth difference between two adjacent surfaces which a person can detect purely by using their stereoscopic vision. This stereo threshold is largely independent of viewing distance when expressed in angular terms [1]. It is usually expressed in seconds of arc (1 arcsec = 1/3600 deg), and is often referred to as stereoacuity.

Stereoacuity is clinically important because stereoscopic vision is considered the "gold standard" of binocular vision [2], requiring good vision in both eyes, good oculomotor control as well as cortical neurons to combine the two eyes' inputs and extract disparity [3]. Accordingly, stereoacuity is a primary or secondary outcome measure in interventions for strabismus and amblyopia [2,4–7], and is routinely measured when children are referred to eye clinics with these conditions.

Several stereo tests are commonly used in the clinic. These include the near Frisby, Frisby-Davis Distance, Lang, TNO, Randot and Preschool Randot stereotests, each with their own properties and (dis)advantages[3,8–10]. Of these, the Randot family of stereotests produced by the Stereo Optical Company (stereooptical.com) are the mostly widely used. The Randot stereotest is the clinical stereotest most often used in the USA and Canada [11], while a PubMed search indicates that Randot tests are also one of the most commonly used for research (the search "testname"[Title/Abstract] AND ((stereo*) OR stereopsis OR amblyopia OR binocular OR strabismus" on 26[th] March 2019 returned 165 results for testname ="Randot", as opposed to 252 for "Titmus", 143 for "TNO", 88 for "Frisby", 83 for "Lang").

In using any stereotest, it is important to know

a. testability, i.e. how many children at each age have the cognitive and other capacity to obtain a meaningful measure on the test.

b. the normative data, i.e. the distribution of values expected at different age-ranges for visually normal individuals.

c. the test/retest reliability, effectively a measure of the "error" on the test. This is particularly important where one wishes to track changes in stereoacuity over time, e.g. as a result of treatment. One needs to know when a given difference in scores reflects real change, and when it is consistent with the measurement error.

d. the sensitivity and specificity with which the test detects binocular visual problems. This is important since stereotests are sometimes included in screening programmes. A child who fails a stereotest will usually be referred for further investigation for a binocular visual problem such as strabismus or amblyopia, while a child who passes the stereotest (and other tests) may be assumed to have no need of referral. Thus it is important to know how many children can be expected to be referred unnecessarily on this basis (the false positive rate, or 1 –specificity) and how many children with binocular visual problems will be missed (the false negative rate, or 1 –sensitivity).

Various studies, summarised in Table 1, have assessed these for the Randot Preschool Stereotest[8,12–19]. However, no study has assessed all of them together, and estimates of reliability, sensitivity and specificity are not available for all age ranges. In this paper, we report these values for the Randot Preschool Stereotest, from a cohort of around a thousand children tested in 2016 in North East England. Comparing them to results from previously published studies, we find generally good agreement, confirming the consistency of the Randot Preschool in different populations. A specific contribution of our paper is a set of equations describing the probability of obtaining a given Randot Preschool stereoacuity as a function of age in years, rather than a simple mean and standard deviation, for visually normal children.

## Methods

### Comparisons with previous studies

To compare our results with previous studies, we conducted a Pubmed search for "Preschool Randot", on 6[th] March 2019. This returned 85 citations, which we reviewed manually to find those which contained relevant data. We excluded any studies which contained data solely for ages over 10 years. For normative values, sensitivity and specificity, we also excluded any studies reporting values solely in a clinical population. We did include reliability measures assessed in clinical populations of the relevant ages. The results of our analysis are summarised in Table 1.

### Participants

A total of 1005 children (488 boys, 517 girls) participated in the study. They were aged between 2 and 12 years old (numbers in each age-group are provided in the Results sections below). The children were recruited through local primary schools, preschools, nurseries, personal contacts, and at local science centres. Testing took place in schools, nurseries, at Newcastle University and at local science centres in the city of Newcastle upon Tyne in North East England. Children were targeted in four UK school year-groups: Nursery (2- and 3-year-olds), Reception (4- and 5-year-olds), Year 2 (6- and 7-year-olds), Year 6 (10- and 11-year-olds). UK school years include children born from September to August of the following year. The study included one 9-year-old who was grouped with the 10- and 11-year-olds. For the Reception year group, our study was combined with the routine Orthoptic School Vision Screening programme, and only children who participated in the screening were eligible to participate. Participation in the Orthoptic School Vision Screening assured that all children were screened for visual problems and the recommended referral pathway was followed. For the other age groups, all children within the targeted ages were eligible to participate. For the three studies reported below (reliability, validity, and normative data), subsamples were used. The criteria and characteristics of the samples are detailed below in the relevant sections.

### Ethics

The study protocol was compliant with the Declaration of Helsinki and was approved by the Ethics Committee of the Newcastle University Faculty of Medical Sciences (approval number 01078). All parents received an information leaflet about the study. For most testing at schools and nurseries, we used opt-out consent, where parents could return a form withdrawing their child from participation. Opt-out consent was approved by our ethics committee in order to ensure a representative sample [21,22]. If requested by the school or nursery and for testing sessions at Newcastle University and local science centres an opt-in consent procedure was used. Children were always asked for oral or non-verbal assent at the time of testing. Parents

**Table 1. Summary of results from this and previous studies.** The study is identified in the "Reference" column; "." indicates this study (values in red). Numbers of children for our study refer to testability and are smaller for other columns; see relevant sections for details. Values in italics were inferred from the data provided in the paper, e.g. by reading off values from figures and/or using the formulae in Table 2. Note that Yang et al[15] claim to provide sensitivity and specificity for the Randot Preschool, but this appears to be incorrect. First, their sample is described as "100 normal children without ocular disease" and specifically excluded children with strabismus or amblyopia, so it is not clear who the "true positives" would be. Second, the denominators used in the calculation of sensitivity and specificity are the number of children in each age-group, not the number of children passing/failing the test. Third, the sensitivity values are over 90% in all age-groups, far higher than any other report. Thus, we have excluded their data.

| Age in years | Reference | Number of children | Testability (% who understand & cooperate) | Normative data (i.e. in children with apparently normal vision) | | | | | | Reliability | | | Validity | |
| | | | | % of testable scoring nil | Stereo thresholds in arcsec | | | Stereo log-thresholds in log₁₀ arcsec | | 95% coefficient of repeatability (1.96 SD of differences[20] in log₁₀arcsec) | | Pearson correlation | Sensitivity \ True Positive % | Specificitiy \ True Negative % |
| | | | | | Mean | Median | SD | Mean | SD | Estimate | 95% CI | | | |
| 2 | . | 49 | 65 | 14 | 382$^c$ | 400 | 287 | 2.44 | 0.38 | | | | | |
| | [14] | | 33 | | | | | | | | | | | |
| | [15] | 19 | 47 | | 332 | | | | | | | | | |
| | [16] | 411 | 31 | | | | | | | | | | | |
| | [17] | 130 | 32 | | | | | | | | | | | |
| | [13] | 400$^h$ | | 2 | 216 | | | *2.17* | *0.37* | | | | | |
| 3 | . | 150 | 92 | 7 | 214$^c$ | 100 | 234 | 2.13 | 0.40 | 0.71 | 0.42–1.0 | 0.72 | 14$^j$ | 91$^j$ |
| | [14] | | 73 | | | | | | | | | | | |
| | [15] | 34 | 85 | | 135 | | | | | | | | | |
| | [16] | 366 | 67 | | | | | | | | | | | |
| | [17] | 287 | 75 | | | | | | | | | | | |
| | [12] | 138 | | | 100$^a$ | | | *150* | *1.74* | *0.47* | | | | |
| | [13] | 1606$^h$ | | 3 | 114 | | | *1.92* | *0.35* | | | | 27$^h$ | 99$^h$ |
| 4 | . | 161 | 99 | 2 | 134$^c$ | 100 | 148 | 1.97 | 0.34 | 0.82 | 0.60–1.03 | 0.05 | | |
| | [14] | | 96 | | | | | | | | | | | |
| | [15] | 25 | 96 | | 71 | | | | | | | | | |
| | [16] | 365 | 88 | | | | | | | | | | | |
| | [17] | 297 | 96 | | | | | | | | | | | |
| | [12] | 217 | | | 100$^a$ | | 50 | *1.95* | *0.21* | | | | | |
| | [13] | 400$^h$ | | 1 | 94 | | | *1.80* | *0.38* | | | | | |
| | [18] | 100$^e$ | | | | | | | | 0.64$^e$ | | 0.97$^e$ | | |
| 5 | . | 101 | 99 | 5 | 87$^c$ | 60 | 76 | 1.84 | 0.27 | 0.64 | 0.42–0.87 | 0.59 | 60$^j$ | 98$^j$ |
| | [14] | | 98 | | | | | | | | | | | |
| | [15] | 22 | 95 | | 51 | | | | | | | | | |
| | [16] | 373 | 95 | | | | | | | | | | | |
| | [17] | 300 | 98 | | | | | | | | | | | |
| | [12] | 104 | | | 60$^a$ | | 70 | *1.59* | *0.40* | | | | | |
| | [13] | 400$^h$ | | 0 | 71 | | | *1.69* | *0.38* | | | | | |
| 6–7 | . | 256 | 100 | 1 | 91$^c$ | 40 | 120 | 1.80 | 0.31 | 0.56 | 0.42–0.70 | 0.72 | 31$^j$ | 99$^j$ |
| | [12] | 46 | | | 60$^{ab}$ | | 20$^b$ | *1.76$^b$* | *0.14$^b$* | | | | | |

*(Continued)*

**Table 1.** (Continued)

| Age in years | Reference | Number of children | Testability (% who understand & cooperate) | Normative data (i.e. in children with apparently normal vision) | | | | | | Reliability | | | Validity | |
|---|---|---|---|---|---|---|---|---|---|---|---|---|---|---|
| | | | | % of testable scoring nil | Stereo thresholds in arcsec | | | Stereo log-thresholds in $\log_{10}$ arcsec | | 95% coefficient of repeatability (1.96 SD of differences[20] in $\log_{10}$arcsec) | | Pearson correlation | Sensitivity \ True Positive % | Specificity \ True Negative % |
| | | | | | Mean | Median | SD | Mean | SD | Estimate | 95% CI | | | |
| 10–11 | . | 195 | 100 | 1 | 63[c] | 40 | 93 | 1.69 | 0.23 | 0.28 | 0.20–0.36 | 0.86 | 26[j] | 99[j] |
| | [12] | 56[i] | | | 40[ai] | 40 | 10[i] | | | | | | 24[g] | |
| | [8] | 19 | | | | | | | | 0.60[d] | 0.35–0.85 | | | |
| | [19] | 47[f] | | | | | | | | 0.23[f] | 0.12–0.35 | | | |

a: Study [12] states means are "rounded to the next larger disparity level available in the Randot Preschool", but this is not consistent with the fact that in some age-groups the mean is given as 40, the best available score, and yet the SD is non-zero, meaning that some children must have scored worse than 40. In this case the pre-rounding mean must have been >40 and so "rounding to the next larger disparity level" would have given a rounded mean of 60. Means may have been rounded to the closest available level.

b. Data from 46 children aged 6 years.

c: We calculated means for non-stereoblind children only, which would reduce our estimates compared to those including all children.

d. For 19 children aged 7–18.

e. For 100 children aged 2–12, but mainly aged around 4. This cohort included 75 clinical patients and 31 out of the 100 were stereoblind, thus ensuring perfect agreement (fail both times).

f. For 47 participants with microtropia, heterophoria or orthophoria, aged 3–80 years, around half children.

g: For 242 patients with amblyogenic conditions, aged 3–18 years, counting 800 arcsec as a fail (57/242 failed).

h: For 1606 children aged 2–5 years, screening for strabismus, counting 800 arcsec as a fail. Sensitivity was 24% screening for amblyopia, and 9% screening for anisometropia; specificity was similarly high for all conditions.

i: Data from 56 children aged 9 and 10 years.

j: Our sensitivity/specificity analysis used larger age-groups, see relevant section.

**Table 2. How to convert between the mean, median, mode and standard deviation of log-thresholds and thresholds, assuming that log-thresholds are distributed normally (Fig 1).**

| Quantity | Units | Symbol | Formulae to convert between these |
|---|---|---|---|
| Mean of log-thresholds | $\log_{10}$ arcsec | M | $M = \log_{10}\mu - \frac{1}{2}\log_{10}\left[1+\left(\frac{\sigma}{\mu}\right)^2\right]$ |
| Standard deviation of log-thresholds | $\log_{10}$ arcsec | S | $S = \sqrt{\frac{\log_{10}\left[1+\left(\frac{\sigma}{\mu}\right)^2\right]}{\ln 10}}$ |
| Median of log-thresholds | $\log_{10}$ arcsec | | $= M$ |
| Mode of log-thresholds | $\log_{10}$ arcsec | | $= M$ |
| Mean of thresholds | arcsec | μ | $\mu = 10^M \exp(0.5(S\ln 10)^2)$ |
| Standard deviation of thresholds | arcsec | σ | $\sigma = \mu\sqrt{\exp((S\ln 10)^2)-1}$ |
| Median of thresholds | arcsec | | $= 10^M = \mu\left[1+\left(\frac{\sigma}{\mu}\right)^2\right]^{-0.5}$ |
| Mode of thresholds | arcsec | | $= 10^{M-S^2\ln 10} = \mu\left[1+\left(\frac{\sigma}{\mu}\right)^2\right]^{-1.5}$ |

and children were informed about the results on standard vision screening test (Visual Acuity and Cover Test) and referred to an optometrist or orthoptists when failing either of these tests.

## Data analysis

Data analysis and statistics were carried out using R (version 3.5.2, "Eggshell Igloo") in Rstudio (version 1.1.463), https://CRAN.R-project.org/. The R data files along with R markdown code to carry out all analysis and figures for this paper are available at https://doi.org/10.25405/data.ncl.9755045.

## Study design and procedures

To evaluate validity and collect a normative data sample, we assessed visual impairments via a questionnaire, visual acuity with a Crowded logMAR test, performed a Cover Test, and a Preschool Randot stereoacuity test. To evaluate reliability a subsample of children was requested to participate in a second session. The second session only included the Randot Preschool stereotest.

**Vision questionnaire.** Parents of participating children were asked to provide information about their child's eye sight: whether they needed glasses for near and / or far vision tasks, whether they were receiving patching of atropine treatment for amblyopia, and to report any other vision problems. In the case of schools and nurseries, this questionnaire was sent home with the child and we requested its return completed. Questionnaires were returned for around half of children (numbers specified in Results sections). To avoid sample bias, we did not exclude children for whom questionnaires were not returned [21,22]. If indicated on the questionnaire, we asked children to wear their glasses during testing. The full questionnaire is available in the Supplementary Material.

**Visual acuity.** Visual acuity was measured in participants tested in non-nursery settings (thus in almost all participants aged 4 years and over) with the Keeler Crowded LogMAR visual acuity test (Keeler Ltd, UK), which is the standard visual acuity test used across the UK [23]. In this test, participants identify or match letters of various sizes presented at a distance of 3 meters with one eye covered. First a screening card is presented to the child and the size of the letters is reduced until an error is made. Once the child answers incorrectly, the examiner starts with the test card two sizes above the last correctly identified letter. If the child is able to identify 2 or more letters on a line, then the next test card is presented. The examiner proceeds until the child is unable to correctly identify 2 letters or more out of the 4 on a line, then the examiner returns to the size above and completes that line. If the 0.800 letter was not seen at 3 meters, our protocol specified that the examiner would walk closer, adding log units to correct for the change in distance. However, all children examined scored at least 0.75 logMAR in both eyes, so this protocol was not used.

Virtually all Reception-year children (4 and 5-year-olds) in our study had their acuity measured as part of the regional Orthoptic School Vision Screening programme. In their protocol, no threshold visual acuity was obtained and the best possible acuity achievable was 0.2 logMAR. Children with a visual acuity above this value were referred to an optician or orthoptist.

Visual acuity measurement was not attempted in nursery settings, as the clinical co-authors advised that in their experience it was not feasible to obtain reliable results with children of this age (2 or 3 years) in such a setting in the available time.

**Cover test.** A cover test was performed to detect manifest strabismus [24]. In this test one eye in turn is covered with an occluder for a short moment while the participant fixates on a near object or penlight, or a distant object. The cover is then briefly removed and the eyes are

observed to see if they move as the occluded eye acquires fixation of the test object. Movement of the unoccluded eye indicates heterotropia. A cover test was performed at near and at distance. The near cover test was always performed at 33 centimetres. The distance for the distance cover test depended on the size of the room that was available. For 42 children the distance for the distance cover test was not recorded. For the other children, the target for the distance cover test was shown at 3 to 6.9 meters away (mean = 4.9, SD = 1.2). Both visual acuity and cover tests were carried out by a qualified orthoptist, either from the study team (SKR) or by an orthoptist from the Newcastle upon Tyne Orthoptic School Vision Screening programme that runs for Reception year group children (4–5 years old).

**Randot Preschool stereotest.** Researchers administered the Randot Preschool Stereoacuity Test. The test consists of 3 pages. At the left-hand side of each page black-and-white silhouettes of everyday objects are presented. The right-hand side shows random dot patterns. In each set of four random-dot patterns one contains no object (is flat), while the remaining contain disparity-defined objects matching one of the silhouette objects presented on the left. The objects are only visible when wearing 3D polarized glasses. The participant has to identify the object in each random dot pattern or point to the matching object on the left page. The available levels are 800, 400, 200, 100, 60 and 40 arcsec. The test distance for Randot Preschool is not specified in the test's own manual, nor in the test protocol described on the Pediatric Eye Disease Investigation Group (www.pedig.net), but we followed previous authors[8,12,13] in performing it at 40cm. Stereopsis was tested at 800 arcsec, 400 arcsec, 200 arcsec, 100 arcsec, 60 arcsec, and 40 arcsec, in that order, following a non-stereo pre-test to check understanding and cooperation. Note that we followed the protocol used in the PEDIG studies rather than that supplied with the test (ATS Miscellaneous Testing Procedures Manual downloaded from www.pedig.net). A lower disparity was shown only if the child could identify at least 2 out of 4 shapes correctly at the previous level. The final score was calculated as the lowest level measured at which 2 of more shapes were correctly identified.

**Converting between threshold and log threshold.** We prefer to work in terms of log-threshold (specifically, the common or decadic logarithm of the threshold in arcsec, measured in $\log_{10}$ arcsec), since the distribution of log-threshold is closer to normal than the distribution of threshold itself [25–28]. Other workers in the field have reported statistics on the threshold itself.

This raises problems when comparing results, since the logarithm of the mean threshold is not the same as the mean of the log-threshold, as illustrated in Fig 1. However, if we assume that log-threshold is indeed distributed normally, then it is possible to derive formulae for converting between statistics on the thresholds themselves and on the log-thresholds. These formulae are provided in Table 2. Note that they require us to know two statistics about the distribution, e.g. both the mean and the SD. In this way, we were able to estimate mean and SD of the log-threshold for previous studies which reported the mean and SD of the threshold in arcsec (Table 1). This was not possible for studies which reported only the mean threshold.

## Results and discussion

### Selection bias with questionnaire

Most of our children were recruited via opt-out consent. Consent forms and study questionnaires were sent home with the children, but if they were not returned, children were still included in the study. We justified this procedure by the non-invasive nature of the study and the importance of avoiding selection bias. Our data-set permitted us to examine the likely effect of any such bias, by comparing children for whom questionnaires were or were not returned.

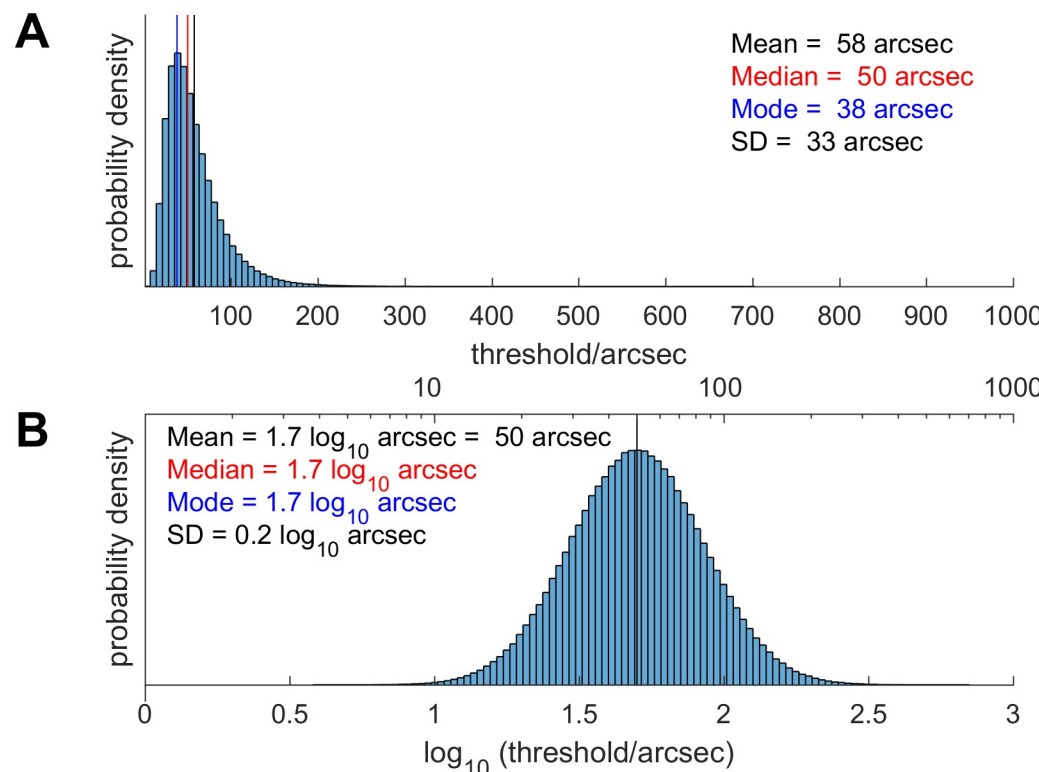

**Fig 1.** Theoretical distribution of stereo thresholds in arcsec (A), assuming that the distribution of log-thresholds is normal (B).The mean, median and mode of the log-thresholds are all equal (1.7 $\log_{10}$ arcsec corresponding to 50 arcsec), but the mean of the thresholds is higher (58 arcsec) and the mode is lower (38 arcsec). The conversion relating the mean, medians etc of these two distributions uses the formulae given in Table 2. The values chosen for the mean and SD of the log-distribution do not matter for the present purpose of illustration, but were taken from the values we obtained in 10 and 11-year-olds.

## Sample

In this section, we used our whole cohort of 1005 children.

**Results.** Overall, questionnaires were returned for 54% of our children. Questionnaires were much more likely to be returned for younger children (e.g. 83% of two-year-olds vs 44% of 11-year-olds; p = 0.0001, logistic regression on age). There was no significant difference by sex. In every age-group, the mean visual acuity and stereoacuity were lower (i.e. better) in children for whom questionnaires were returned (Fig 2). This difference was significant in both cases (visual acuity: regression of visual acuity on log age with questionnaire as a categorical factor, *p* = 0.0008; stereoacuity: ordinal logistic regression (see below) with log age and questionnaire as factors, *p* = 0.04). For stereoacuity, the advantage of having parents who returned a questionnaire was equivalent to being 0.6 years older.

**Discussion.** We had imagined that parents might be more likely to return questionnaires for children who had a diagnosed eye condition such as amblyopia. With such an effect, vision would have tended to be worse in children with questionnaires. In fact, we find the opposite effect. The possible cause is beyond the scope of our study; nevertheless these results suggest that our sample would have been biased towards children with better vision if we had used opt-in consent.

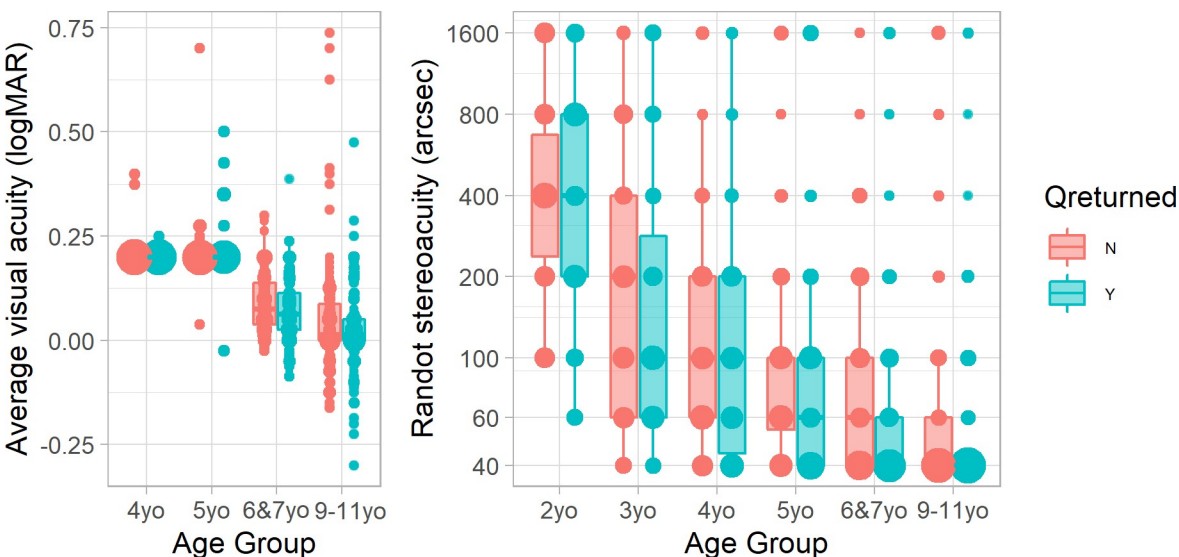

**Fig 2.** (A) Visual acuity (average of left and right eyes) and (B) stereoacuity, by age-group, for children for whom a questionnaire was (blue, right) or was not (red, left) returned. Size of symbols show proportion obtaining the given score; Tukey box and whisker plots show interquartile ranges. Note that most children aged under 6 only underwent a screening acuity test, on which the best score was 0.2 logMAR.

### Testability with Randot Preschool stereotest

**Sample.** For the testability analysis, we included all 1005 children. The data is provided in data-file RandotPreschoolTestability.RData and the analysis in this section can be recreated by running R markdown file AnalyseTestabilityData.Rmd (see Supplementary Material).

**Results.** Fig 3 and Table 3 shows the percentage of children who were testable in each age group. Testability rose from 65% in two-year-olds and 92% in three-year olds, to virtually 100% in older children. Slightly more girls than boys were testable in the youngest age-group, but this difference was not significant. Reasons for non-testability included failing the non-stereo pre-test (which requires naming simple luminance-defined black-on-white shapes), not being willing to wear the 3D glasses (including one two-year-old who burst into tears after putting them on!), and not understanding what they were being asked to do.

**Discussion.** Our results agree with previous studies in finding virtually 100% testability in children aged 4 and up [14–17,29], but we find substantially higher testability in two-year-olds than previous studies: 65% compared to 31% [16], 32% [17], 33% [14], or 47% [15].

### Test/retest reliability of Randot Preschool stereotest

**Sample.** A subset of children aged three and older were retested on Preschool Randot in a separate session within three weeks of the first test (max 21, mean 16 days apart). We compared results on the two sessions to calculate the test/retest reliability. For this analysis, we did not exclude any children because of visual conditions, but we did exclude children who were recorded as having worn optical correction on one session but not on the other, since we did not want to confound poor test reliability with changes in optical correction. We also excluded children who did not understand the test the first time, as we wanted to understand repeatability of results independent of changes in understanding. In total, this sample consisted of 182 children from 3 to 11 years old.

**Results.** We initially looked at the reliability of a binary pass/fail classification. We define "failing" as not passing the 800 arcsec level. In the ideal case where the test is 100% reliable and

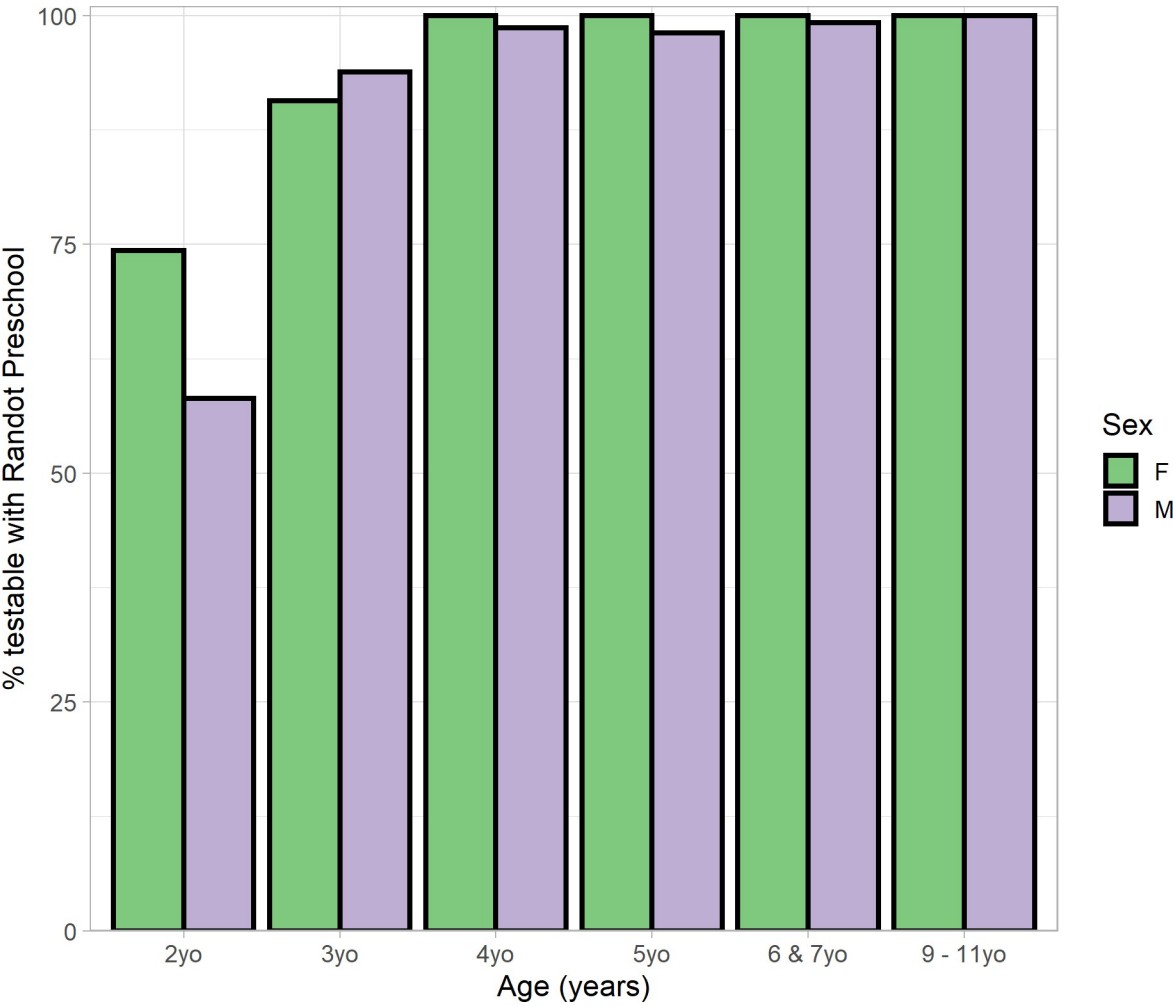

**Fig 3. Testability on the Randot Preschool, by age-group and sex.**

independent of cognitive effects or motivation, children should either pass the test on both sessions or fail on both. P

Fig 4 shows the proportion of children in each situation, classified by their age-group on the first session. Overall 96% of children passed Preschool Randot both times. Only 1% (2 children) failed on the second session having previously passed, and one of these, an 11-year-old, had obtained 800 arcsec on the first session. Thus, the Randot Preschool is extremely

**Table 3. Testability on the Randot Preschool, by age-group and sex.**

| Age Group | N children | % testable | N girls | % testable | N boys | % testable |
|---|---|---|---|---|---|---|
| 2yo | 78 | 65 | 35 | 74 | 43 | 58 |
| 3yo | 167 | 92 | 86 | 91 | 81 | 94 |
| 4yo | 171 | 99 | 96 | 100 | 75 | 99 |
| 5yo | 106 | 99 | 53 | 100 | 53 | 98 |
| 6 & 7yo | 267 | 100 | 139 | 100 | 128 | 99 |
| 9–11yo | 216 | 100 | 108 | 100 | 108 | 100 |

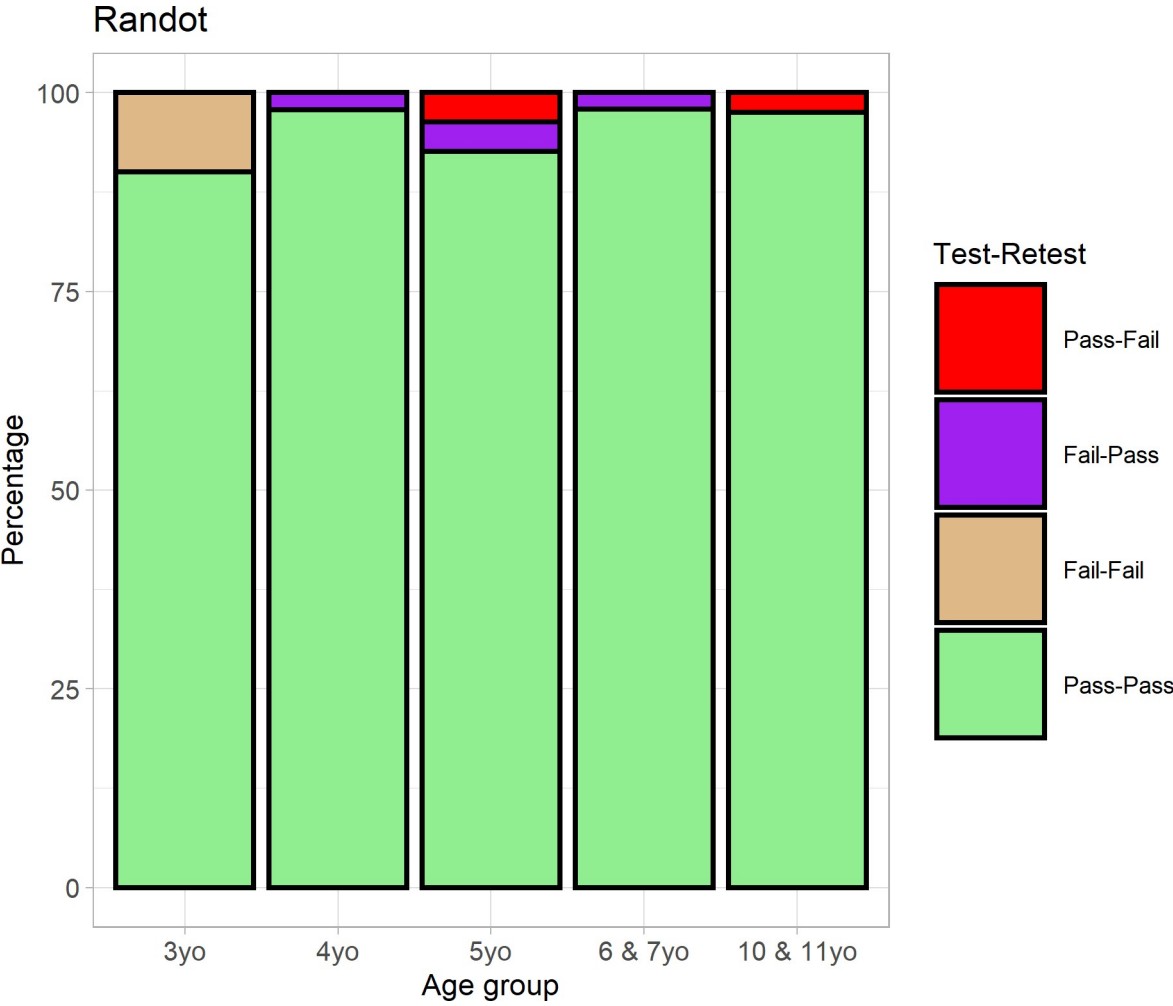

**Fig 4. 187 children were tested in two sessions (22 three-year-olds, 50 four-year-olds,27 five-year-olds, 48 aged 6 or 7 and 40 aged 10 and 11) and on each session we classify them as passing/failing the stereotest.** The light-colored bars (green & beige) show the high level of agreement.

consistent at classifying children into those who do versus those who do not have any demonstrable stereo vision.

To assess agreement in more detail, we first examined the correlation between scores on the two tests. The Spearman correlation coefficient, which examines the ranking of scores, was 0.59. To compute the Pearson correlation, following previous workers[18], we first replaced a "nil" score with a notional level of 1600 arcsec, i.e. one log-level up from the highest available score of 800 arcsec. The Pearson correlation coefficient between the log-thresholds on the two sessions was 0.62. Both correlations were extremely significant ($p < 10^{-10}$).

We then carried out a Bland-Altman analysis, shown in Fig 5. Following previous workers [8,18,19], we analysed log-thresholds rather than thresholds, since these are closer to normally distributed. The mean difference between results on the two sessions was -0.114 log-arcsec or a factor of 0.77, and this was significantly different from zero ($p < 10^{-5}$). Thus, children tended to obtain a better score on the second session, presumably due to practice effects.

Since this improvement is relatively small compared with the variability between the two sessions (Fig 5), we will neglect it in computing the reliability. We quantify reliability using the

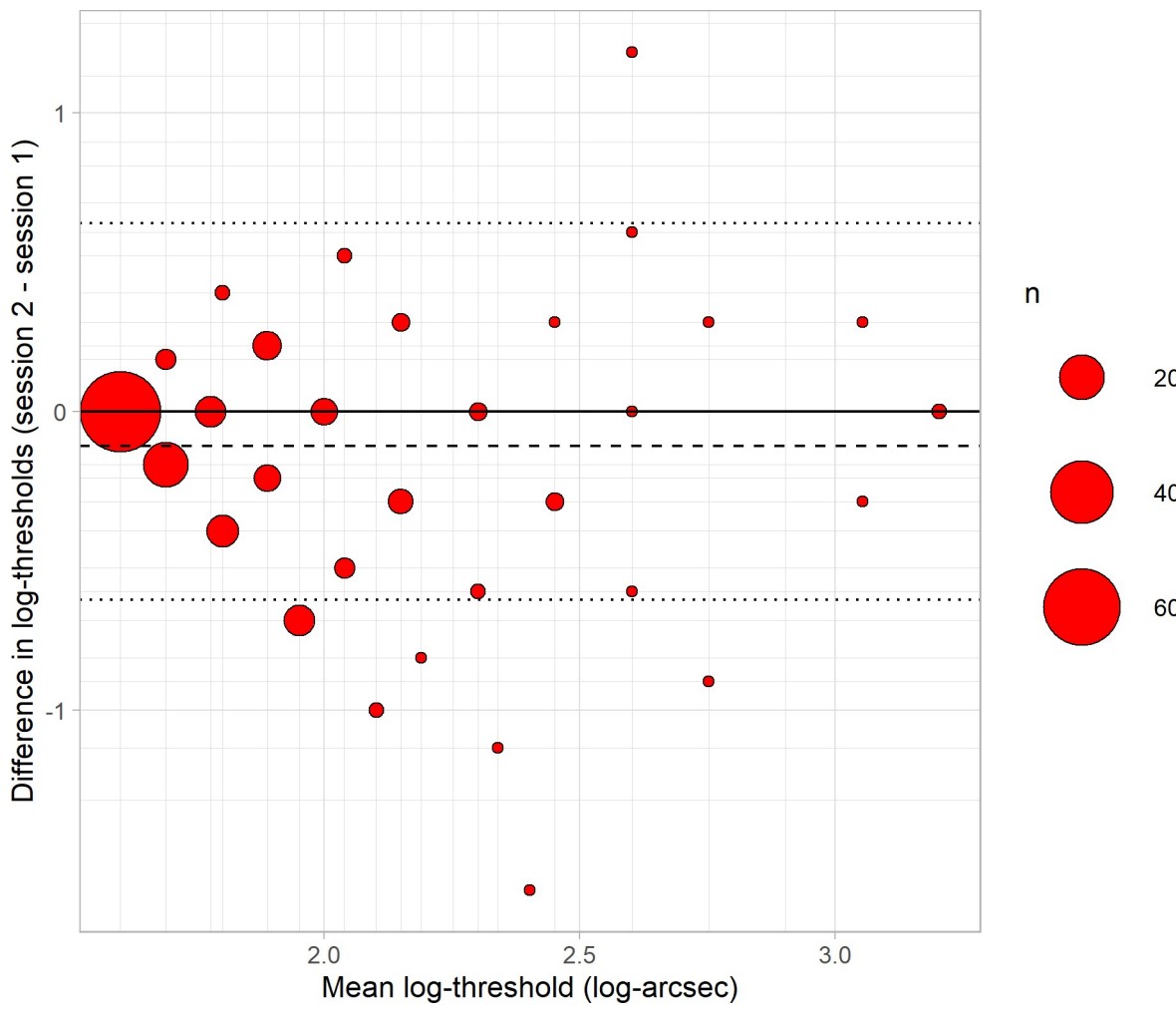

**Fig 5. Difference between log-thresholds in the two sessions (second minus first) plotted against means, for 187 children aged 3–11 years.** The size of the symbol indicates the number of children (see legend). Positive differences indicate worse performance on re-test, negative differences indicate better; no change is indicated by the solid black line. Dashed line shows the mean of the differences, dotted lines show the 95% limits of reliability about 0. The minor grid lines indicate the possible mean and differences given the available values on the Randot Preschool.

Bland-Altman 95% limit of agreement, where a value of $L$ means that one can be 95% confident that the result of a second test will lie within ± $L$ of the first. For a normal distribution, this corresponds to 1.96 times the standard deviation of the differences. In our Preschool Randot data, this gives $L = 0.63 \log_{10}$ arcsec, corresponding to a factor of 4.3 in thresholds. For example, if a children scores 200 arcsec in the first session, their score in the second session could be between 50 arcsec and 800 arcsec, without any change in their binocular vision.

One can also compute the 95% confidence interval on the estimate $L$. We follow Bland & Altman's (1986) recipe for this, estimating the standard error on the limit of agreement as $\sqrt{(3s^2/n)}$, where $s$ is the standard deviation of the differences and $n$ is the sample size. We then estimate the 95% confidence interval as the original estimate ± t times the standard error, where t is the t-statistic corresponding to the 95% confidence interval (1.96 for an infinite sample). In this way we estimate a 95% confidence interval of 0.55 to 0.71 $\log_{10}$ arcsec (factors of 3.5 to 5.2).

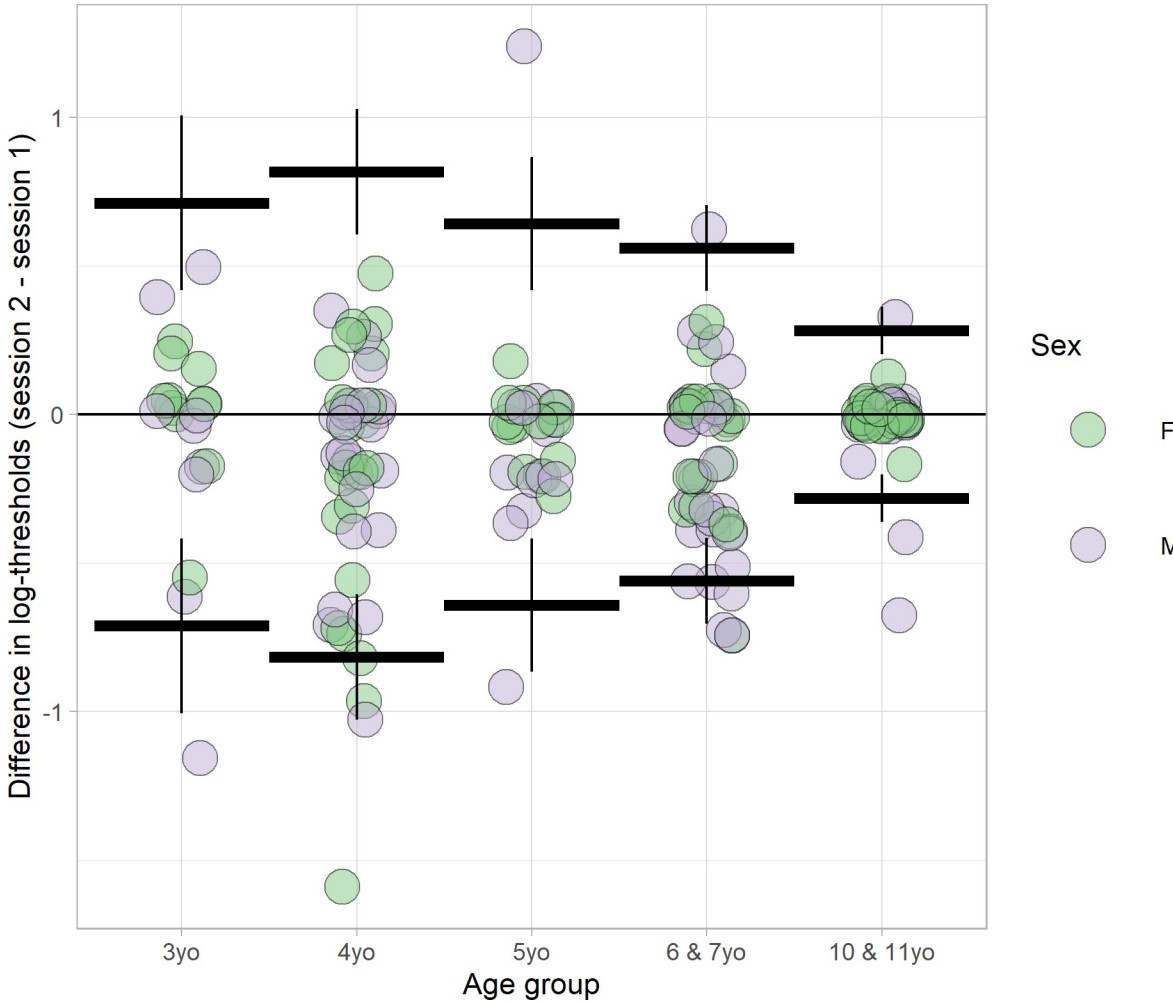

**Fig 6. Differences in Preschool Randot thresholds (second session minus first session) are plotted by age-group for 182 children.** The thick horizontal lines show the 95% limits of reliability for the 5 age-groups; the vertical lines show the estimated 95% confidence interval on this limit.

Fig 6 shows the reliability as a function of age. Values are given in Table 4. Reliability is similar between ages 3–7, but better in the oldest age-group, 10 & 11 year old. A linear regression of absolute difference in log-thresholds against age, with sex as a covariate, revealed a highly significant decrease in absolute difference with age (p<0.058). Absolute differences were slightly higher for boys, though this was not significant (offset = 0.074, p = 0.058).

**Discussion.** Our estimate of the reliability of Preschool Randot is very similar to previous estimates, with the 95% limits of repeatability being ±0.64 $\log_{10}$ arcsec or a factor of 4.3 in threshold. Fawcett & Birch [18], in 102 children aged 2–12 years, obtained exactly the same value as us: 0.64 $\log_{10}$ arcsec. Adams et al[8] report a similar value of ±0.60 $\log_{10}$ arcsec (a factor of 4) with 19 children aged 7–18, while Smith et al obtained a slightly lower value, 0.46 $\log_{10}$ arcsec (a factor of 3), in 47 people aged 3 to 80 years. The differences likely reflect an improvement in reliability with age.

A previous study [18] concluded there was no change in test/retest reliability over the range 3–12 years. However, this conclusion was based on a linear regression and t-test on the differences themselves, rather than the absolute difference. This is in fact testing for a change in the

**Table 4. 95% limits of reliability and 95% confidence interval by age-group.** The low correlations obtained for the 4-year-olds are correct; they are partly because only one child in this age-group scored 1600 arcsec on either test (and this child then scored 40 arcsec on retest), so there was a lower range of values (note the high confidence intervals).

| Age-group | N children | 95% limits of reliability and 95% confidence interval on these | | Correlation coefficients | |
|---|---|---|---|---|---|
| | | In $\log_{10}$ arcsec | As a factor | Spearman | Pearson |
| 3 year olds | 20 | 0.71 (0.42 to 1.01) | 5.15 (2.61 to 10.13) | 0.47 (0.036–0.76) | 0.72 (0.4–0.88) |
| 4 year olds | 47 | 0.82 (0.60 to 1.03) | 6.54 (4.02 to 10.65) | 0.13 (-0.17–0.4) | 0.045 (-0.25–0.33) |
| 5 year olds | 27 | 0.64 (0.42 to 0.87) | 4.38 (2.61 to 7.35) | 0.79 (0.58–0.9) | 0.59 (0.28–0.79) |
| 6 & 7 year olds | 48 | 0.56 (0.42 to 0.70) | 3.63 (2.61 to 5.06) | 0.65 (0.46–0.79) | 0.72 (0.54–0.83) |
| 10 & 11 year olds | 40 | 0.28 (0.20 to 0.36) | 1.91 (1.59 to 2.30) | 0.47 (0.19–0.68) | 0.86 (0.74–0.92) |
| All | 182 | 0.63 (0.55 to 0.71) | 4.27 (3.54 to 5.15) | 0.59 (0.49–0.68) | 0.62 (0.52–0.7) |

bias (i.e. the mean of the differences), rather than in the reliability. An increase in reliability with age would be expected to decrease the variance of the difference in the scores obtained on two sessions, without changing its mean. This is why we did our linear regression on the *absolute values* of the differences. An F-test confirms that there is a highly significant decrease in the variance of differences in log-thresholds with age. In our data, this variance is 2.38 times higher in children under 5 than in those aged 5 and over, and in study [18] the figure is actually 3.00 times higher (data read off from their Fig 3 [18]); $p < 10^{-3}$ for both.

However, this apparent improvement in reliability with age may in fact reflect improved stereo thresholds combined with a floor effect in the scores available. As reported in the section on Normative values, almost all older children obtain the best possible threshold of 40 arcsec on the Randot Preschool. Suppose that the reliability is in fact a factor of 4 at all ages. A young child whose true threshold is 200 arcsec may obtain 400 arcsec on one session and 100 on the next. Yet an older child whose true threshold is 20 arcsec will obtain 40 arcsec on both sessions, simply because a score of 10 arcsec is not available. Reliability will appear higher for the older child but this will be a side-effect of their improved stereo, combined with the available scores. The Randot Preschool was not designed to assess genuine changes in the repeatability of stereo thresholds in general over this age range.

## Normative values of Randot Preschool stereotest

**Sample.** We next investigated the distribution of stereo thresholds obtained with the Randot Preschool in children who, as far as we could tell, had normal vision. To this end, we excluded participants who failed a cover test, or in whom cover test data were not available. We also excluded participants whose parents reported that they were diagnosed or treated for amblyopia or strabismus, or under assessment in an eye clinic for suspected vision problems, or who when tested were not wearing glasses when their parents reported that they needed glasses; but we did not exclude children for whom parental questionnaires were not available. For children aged 4 and over, we also excluded participants for whom visual acuity data were not available, or whose visual acuity was worse than 0.2 logMAR in either eye, or whose interocular acuity difference exceeded 0.2 logMAR. Table 5 summarises the visual acuity of our normative sample. In the 2 and 3 year-olds, who were tested in nurseries, visual acuity data was not available. Our "normative" sample may therefore include some children in this age-range with undiagnosed poor vision. We also excluded any children who were not testable with Randot because they refused to wear the glasses or did not cooperate with the test in another way (see section on Testability for age breakdown). The remaining 826 participants were 402 boys and 424 girls aged between 2.00 and 11.6 years (Fig 7 and Table 5). Analysis code is provided

**Table 5. Visual acuity in the two older age-groups of our normative sample.** All visual acuities are in logMAR, mean ± SD. Children with visual acuity worse than 0.2 logMAR in either eye, or an interocular visual acuity difference >0.2 logMAR, were excluded from the normative sample.

| Age | 6 & 7 years old | 10 & 11 years old |
|---|---|---|
| Number of children | 247 | 190 |
| Monocular acuity in better eye | 0.047 +/- 0.067 | -0.012 +/- 0.084 |
| Monocular acuity averaged across eyes | 0.07 +/- 0.064 | 0.01 +/- 0.081 |
| Interocular acuity difference | 0.0033 +/- 0.062 | 0.011 +/- 0.067 |
| Acuity in left eye | 0.068 +/- 0.073 | 0.0049 +/- 0.087 |
| Acuity in right eye | 0.071 +/- 0.069 | 0.016 +/- 0.088 |

in AnalyseNormativeData.Rmd using data-file RandotPreschoolNormative.RData (Supplementary Material).

**Results.** Fig 8 and Table 6 report the distribution of Randot Preschool scores in each age-group. There is a clear improvement in stereoacuity with age. Additionally, scores are

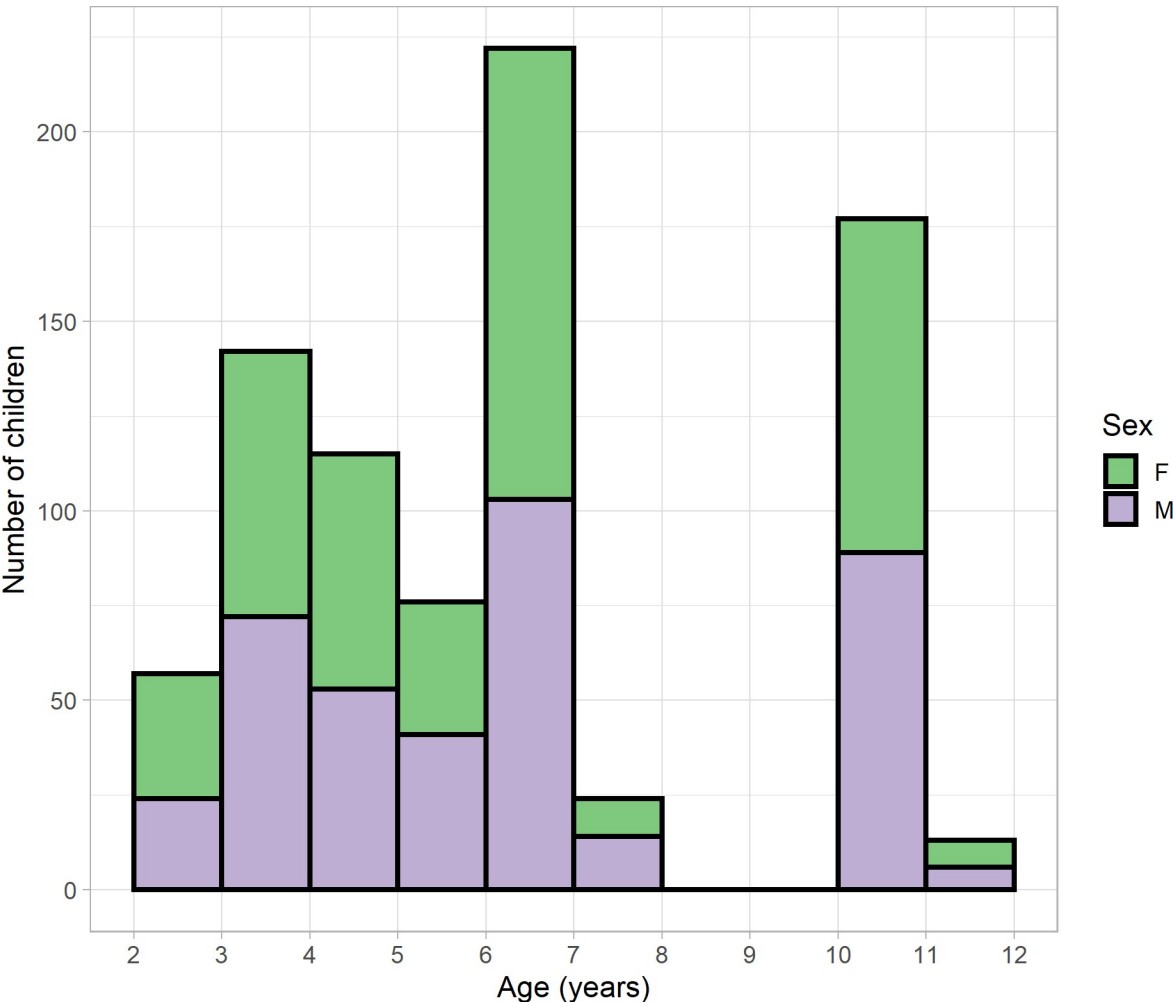

**Fig 7. Age distribution of normative data sample, showing a similar number of boys (mauve, total 465) and girls (green, 487) for each age.**

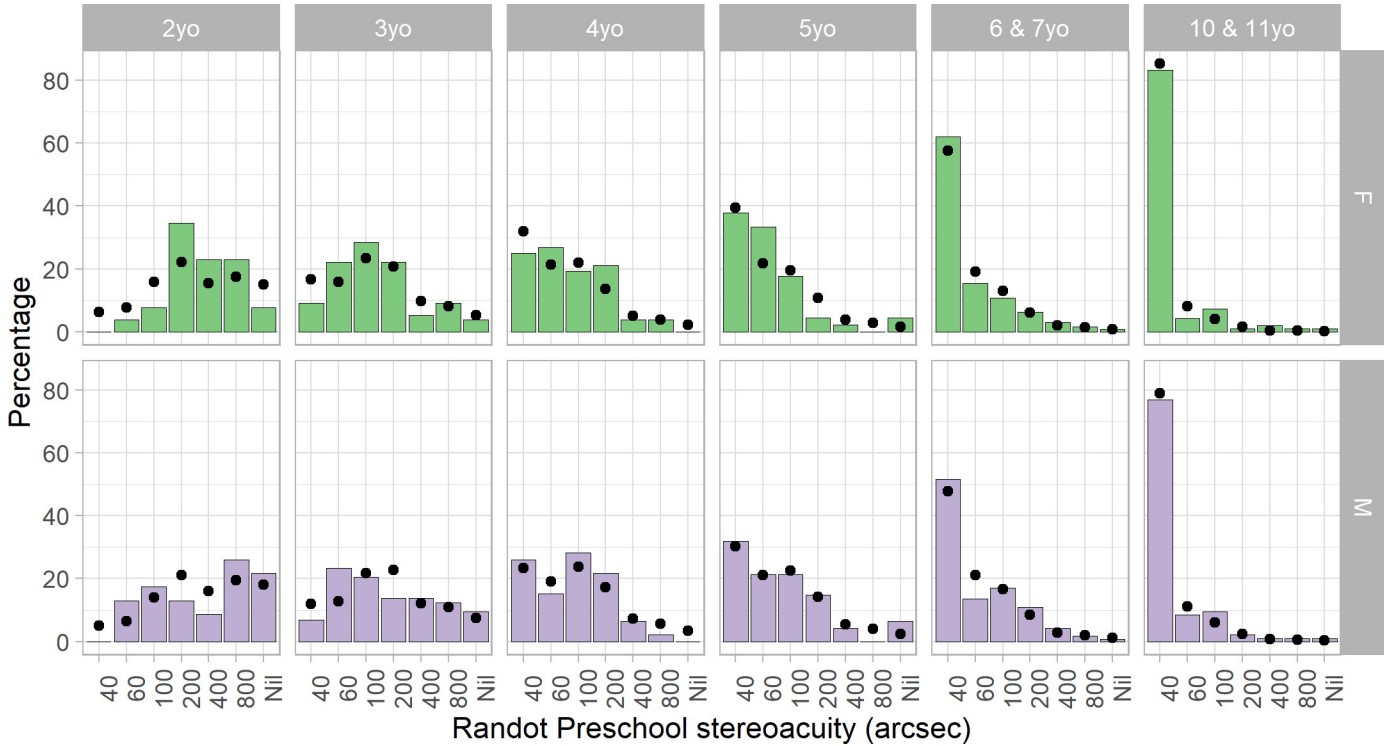

**Fig 8. Distribution of normative Randot Preschool stereo thresholds in different age-groups, for children with normal vision, separated by sex.** In children aged 4 and up, the most common outcome is 40 arcsec, the best score available. Dots show the fit of a descriptive model fitted to this data (Eq 1). For each child in the study, given their age and sex, the model estimated the probability of each possible score. We then averaged these across all children of a given sex and age-group to obtain the points shown. With 8 parameters fitted to the data, the model gives a reasonable account of the data in these 84 bins.

consistently slightly better for girls. To quantify this, we carried out an ordinal logistic regression, using function polr from R package MASS [30]. The main effect of age and sex were both significant ($p < 10^{-10}$ for age and $p = 0.002$ for sex). The better performance of girls was most pronounced in the 2 and 3 year-olds, where visual acuity was not measured, but remained significant even if only children in the two older age-groups (over 5 years) are considered.

**Table 6. Randot Preschool stereo thresholds by age-group, for children with normal vision who were judged as being able to understand and cooperate with the test.** Mean and SDs are reported for those children who could complete at least the 800 arcsec test level. Percentiles (type 1 quantile from Hyndman and Fan (1996)) are for all children, including those who scored "nil". Stereothresholds in the normal range for each age group are marked in green (up to the 75% percentile). "Nil" means unable to perform 800 arcsec plate of Randot Preschool despite passing non-stereo pre-test.

| Age-group | Number tested | % scoring nil | Mean, SD computed on those who could complete at least 800 arcsec | | | | Percentiles computed on all tested | | | | |
|---|---|---|---|---|---|---|---|---|---|---|---|
| | | | Threshold in arcsec | | Log-threshold in $\log_{10}$ arcsec | | Percentile in arcsec | | | | |
| | | | Mean | SD | Mean | SD | 25% | 50% | 75% | 90% | 95% |
| 2yo | 49 | 14. | 382. | 287. | 2.44 | 0.380 | 200 | 400 | 800 | Nil | Nil |
| 3yo | 150 | 6.7 | 214. | 234. | 2.13 | 0.395 | 60 | 100 | 400 | 800 | Nil |
| 4yo | 98 | 0.0 | 134. | 148. | 1.97 | 0.338 | 40 | 100 | 200 | 200 | 400 |
| 5yo | 92 | 5.4 | 87.1 | 76.3 | 1.84 | 0.265 | 40 | 60 | 100 | 200 | Nil |
| 6 & 7yo | 247 | 0.81 | 90.6 | 120. | 1.80 | 0.306 | 40 | 40 | 100 | 200 | 400 |
| 10 & 11yo | 188 | 1.1 | 63.0 | 92.6 | 1.69 | 0.225 | 40 | 40 | 40 | 100 | 200 |
| All | 824 | 3.2 | 126. | 173. | 1.89 | 0.368 | 40 | 60 | 100 | 400 | 800 |

**Table 7. Fitted parameters for the ordinal logistic model (see Eq 1 for how to use).** The negative regression slope β represents the improvement with age; the positive γ represents the poorer performance of boys. The model residual deviance is 2305 compared to 2635 for a model ignoring age and sex (thus with two fewer free parameters).

| Parameter | Estimated value | Significance |
|---|---|---|
| Age regression slope β (per $\log_{10}$ year) | -3.079 | $<10^{-10}$ |
| Effect of being male γ | +0.4152 | 0.002 |
| $\alpha_{40\|60}$ | -5.513 | $<10^{-10}$ |
| $\alpha_{60\|100}$ | -4.623 | $<10^{-10}$ |
| $\alpha_{100\|200}$ | -3.639 | $<10^{-10}$ |
| $\alpha_{200\|400}$ | -2.679 | $<10^{-10}$ |
| $\alpha_{400\|800}$ | -2.007 | $<10^{-10}$ |
| $\alpha_{800\|Nil}$ | -0.9797 | 0.003 |

In the ordinal logistic regression model, the probability of obtaining a stereo threshold equal to or better than a particular Randot Preschool level $L$ is modelled as

$$P(R \leq L) = [1 + \exp(-\alpha_L + \beta A + \gamma M)]^{-1}$$

where $R$ is score on Randot Preschool stereotest, $A$ is log age in $\log_{10}$(years), $M$ is a categorical variable specifying sex ($M = 1$ for males and $M = 0$ for females), $\beta$ is a fitted parameter describing the effect of age $A$, $\gamma$ is a fitted parameter describing the effect of sex, and the six fitted parameters $\alpha_L$ depend on the level $L$ (Table 7). For our data, the 8 fitted parameters (the regression slope for age and gender plus 6 coefficients describing the transitions between the 7 different Randot levels) were as specified in Table 7. The predicted probability of obtaining each Randot Preschool score as a function of age $A$ in years is then:

$$P(R = 40 \text{ arcsec}) = [1 + \exp(-\alpha_{40|60} + \beta A + \gamma M)]^{-1}$$

$$P(R = 60 \text{ arcsec}) = [1 + \exp(-\alpha_{60|100} + \beta A + \gamma M)]^{-1} - [1 + \exp(-\alpha_{40|60} + \beta A + \gamma M)]^{-1}$$

$$P(R = 100 \text{ arcsec}) = [1 + \exp(-\alpha_{100|200} + \beta A + \gamma M)]^{-1} - [1 + \exp(-\alpha_{60|100} + \beta A + \gamma M)]^{-1}$$

$$P(R = 200 \text{ arcsec}) = [1 + \exp(-\alpha_{200|400} + \beta A + \gamma M)]^{-1} - [1 + \exp(-\alpha_{100|200} + \beta A + \gamma M)]^{-1} \text{ Eq 1}$$

$$P(R = 400 \text{ arcsec}) = [1 + \exp(-\alpha_{400|800} + \beta A + \gamma M)]^{-1} - [1 + \exp(-\alpha_{200|400} + \beta A + \gamma M)]^{-1}$$

$$P(R = 800 \text{ arcsec}) = [1 + \exp(-\alpha_{800|Nil} + \beta A + \gamma M)]^{-1} - [1 + \exp(-\alpha_{400|800} + \beta A + \gamma M)]^{-1}$$

$$P(R = Nil) = 1 - [1 + \exp(-\alpha_{800|Nil} + \beta A + \gamma M)]^{-1}$$

**Discussion.** As it was designed to, the Randot Preschool contains a range of test values suitable for assessing threshold stereoacuity in pre-school children (aged 2–5 years). For older children, most visually normal children will obtain the best available score, and so the test is more appropriate as a screening tool.

Normative values from ours and other studies are reported in Table 1. Our values are a little higher than in previous studies [12,13,15]. A potential reason could be that, as described in the Methods, we followed the PEDIG protocol for the Randot Preschool rather than the manufacturer's instructions. This means that we started the stereo part of the test at 800 arcsec and proceeded to smaller disparities, while the manufacturer instructs testers to begin at 200 arcsec and move up or down depending on results. The latter increases the probability of obtaining a good score by chance, and similarly means that a child who ceases responding after a few trials

will obtain a better score (if poor motivation is mistaken for inability to do the task). However, previous studies using the PEDIG [13] or manufacturer's [12] protocols have obtained similar results, suggesting that this is unlikely to be responsible. Perhaps the most plausible reason could be poor acuity in our "normative" sample. Visual acuity was not measured in our 2- and 3-year-olds, so these groups likely included some children with undiagnosed poor vision. However, we also obtain higher scores in the older age-groups, where all children had visual acuity better than 0.2 logMAR in both eyes. Additionally, an unknown fraction of our sample may have not been wearing the correct refractive correction. The previous studies performed more thorough screening for normal vision, e.g. performing cycloplegic refraction and excluding children with anisometropia >1D [13]. However, anisometropia would be expected to affect stereoacuity via an effect on visual acuity, and we still obtain slightly higher scores even with more stringent limits on visual acuity and interocular acuity differences (S1 Table in S1 File).

Surprisingly, we have found a small but significant sex difference in stereoacuity, with girls scoring slightly better (i.e. lower) than boys, especially at younger ages. This has not previously been reported. In adults, female interpupillary distance is smaller than males', meaning that a given depth creates a smaller angular disparity at the retina. Thus, females might be expected to develop sensitivity to smaller disparities, i.e. better stereoacuity. To our knowledge, only one publication has reported an effect of interpupillary distance [31], and this was in the wrong direction (the study used the Frisby stereotest, which uses real depth, so observers with larger interpupillary distance should have experienced more disparity for a given test level; yet in fact thresholds increased with interpupillary distance). Other studies have found no sex difference in stereoacuity in adults [32,33], and also no effect of interpupillary distance [33,34]. Sex differences in interpupillary distance become more pronounced during development [35,36], whereas our sex differences in stereoacuity are more pronounced in the younger age-groups. For these reasons, we think it is unlikely that interpupillary distance accounts for the sex difference. It may be related to sex differences in binocular function reported very early in life [37–39], with stereopsis emerging earlier in girl babies and their vergence responses being more responsive to disparity than boys'. It may also reflect non-visual cognitive/social/developmental sex differences, e.g. in willingness to cooperate. Boys in our cohort also showed slightly lower testability and test/retest reliability (Fig 3, Fig 6), though these differences were not significant. Further work would be needed to establish whether this sex difference in stereoacuity is reproducible and to establish its causes.

**Sensitivity and specificity of Randot Preschool stereotest.** Stereotests are commonly used in visual screening. The aim is to identify children with binocular vision problems by their failing the stereotest. Here, we evaluate the sensitivity and specificity of the Randot Preschool used in this way. *Sensitivity* or *true positive rate* is the proportion of children with binocular vision problems who failed the stereotest. *Specificity* or *true negative rate* is the proportion of patients without binocular vision problems who passed the stereotest (Fig 9).

**Sample.** For this analysis, we included children in whom we were able to perform the Randot Preschool (i.e. they were testable) and the cover test; in children over 4, we also required that visual acuity was measured in both eyes. This left 892 children, for 480 of whom parental questionnaires were available. Since binocular vision problems were rare (only 37 of the 892 children had one of the binocular visual problems defined in the next paragraph), for this analysis we combined the younger age-groups so that we had at least 200 children in each group. The analysis code is available in AnalyseValidityData.Rmd using data in RandotPreschoolValidation.RData.

**Results.** Fig 10 shows the percentage of children scoring in each of the Randot Preschool levels, by age, color-coded to show whether their vision appeared entirely normal (green) or

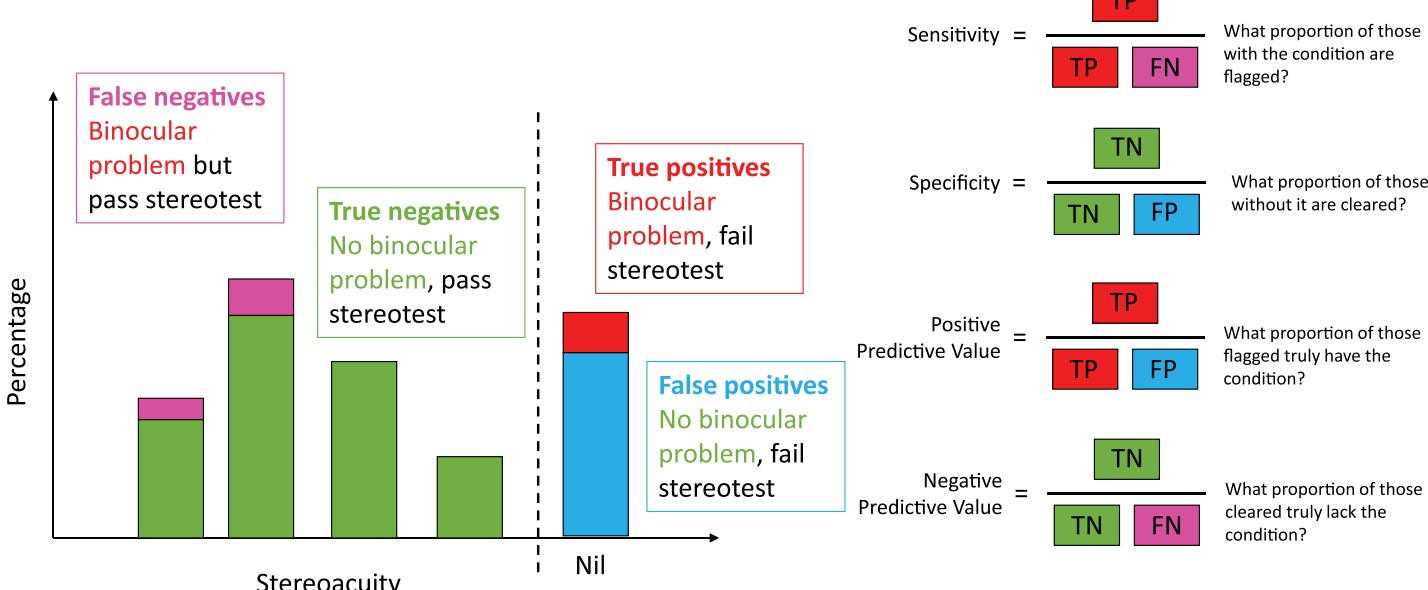

**Fig 9. Definition of sensitivity, specificity, and positive/negative predictive value, in the context of stereotests.**

whether a binocular visual problem was identified. We classified children as having a problem likely to affect binocular vision if (a) their parental questionnaire (if available) clearly indicated one, e.g. "attends eye clinic for lazy eye" (purple, "Parent" in Fig 10); or (b) our cover test showed clear evidence of a problem, e.g. exotropia (red, "CTFAIL"); or (c) their visual acuity in the poorer eye exceeded 0.48 logMAR, the WHO threshold for "moderate visual impairment" (brown, "ModVI"); or (d) they had an interocular visual acuity difference in excess of 0.2 logMAR (gold, "IADonly"). Note that each class includes the ones to its right in the legend. Thus only one child appears in the graph purely because of a large interocular acuity difference (gold), but children who failed the cover test (red) may also have had a large interocular acuity difference. We included all four categories when classifying children as having a binocular vision problem.

Table 8 shows the specificity, sensitivity and positive/negative predicted value for the Randot Preschool stereo test in detecting a binocular vision problem. In all age-groups, the specificity is very high ($>\sim$ 90%) but the sensitivity is poor. The high specificity means that nearly all children without binocular vision problems pass the Randot Preschool. However, the poor sensitivity means that many children with binocular vision problems also pass. The positive predict value is fairly good, especially in older age-groups, meaning that most children over 5 who fail the Randot Preschool do have a binocular visual problem. The numbers in Table 8 reflect our fixed pass level of 800 arcsec, but as is apparent from Fig 10, there is no alternative choice of criterion–even varying it by age–which would permit higher sensitivity while retaining the high specificity. We also explored other definitions of "moderate visual impairment", without finding a better criterion.

A possible concern is that some children for whom visual acuity or parental questionnaire were not available may have had visual problems which we did not detect. They would therefore be erroneously classed as "false positives" if they failed the Randot Preschool, rather than "true positives". We therefore also examined the results when including only children for whom visual acuity and questionnaires were available (S1 Fig in S1 File, S2 Table in S1 File), but this did not change the conclusions. In fact the sensitivity values were even poorer.

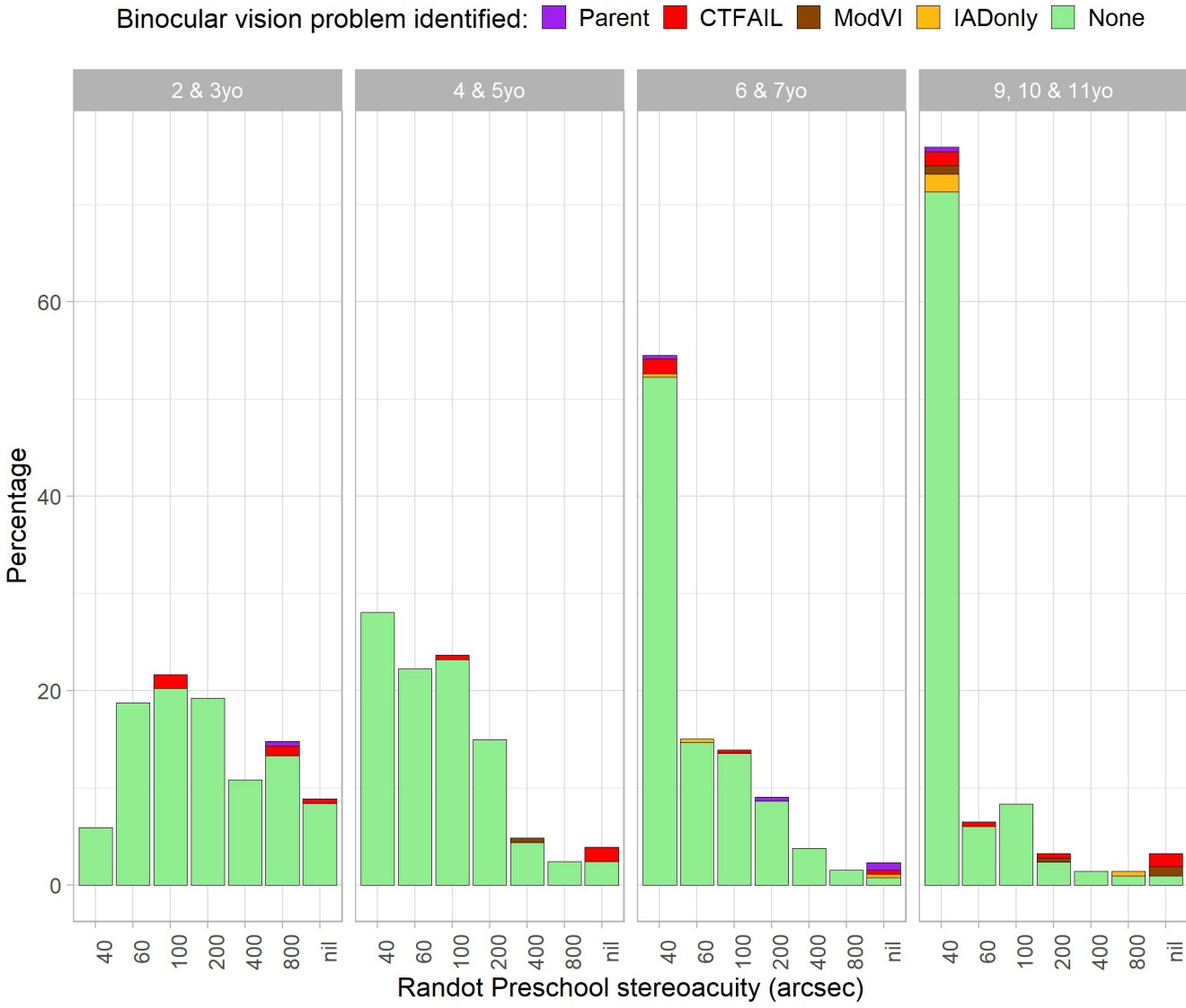

**Fig 10. Randot Preschool stereo thresholds by age-group, color-coded to indicate whether any binocular vision problems were identified.** Parent: parental questionnaire indicated strabismus or amblyopia; CTFAIL: cover test found heterotropia; ModVI: acuity in worse eye >0.48 logMAR; IADonly: interocular acuity difference >0.3 logMAR (but no other abnormality). Note that each type of binocular vision problem includes the ones on the right; e.g. those colored red for 'CTFAIL' may also have had acuity >0.48 logMAR in one eye, but those colored brown for 'ModVI' must have passed the cover test.

**Table 8. Predictive value of Randot Preschool in detecting binocular vision problems, by age-group, taking "pass" as a score of 800 arcsec or lower.** N = number of children, TP = number of true positives (those who had a binocular vision problem and failed the stereotest), FP = number of false positives, TN = number of true negatives, FN = number of false negatives, PPV = positive predictive value, NPV = negative predictive value (see Fig 9).

| | N | TP | FP | TN | FN | Sensitivity | Specificity | PPV | NPV |
|---|---|---|---|---|---|---|---|---|---|
| 2 & 3yo | 203 | 1 | 17 | 179 | 6 | 14 | 91 | 6 | 97 |
| 4 & 5yo | 207 | 3 | 5 | 197 | 2 | 60 | 98 | 38 | 99 |
| 6 & 7yo | 266 | 4 | 2 | 251 | 9 | 31 | 99 | 67 | 97 |
| 9, 10 & 11yo | 216 | 5 | 2 | 195 | 14 | 26 | 99 | 71 | 93 |
| All | 892 | 13 | 26 | 822 | 31 | 30 | 97 | 33 | 96 |

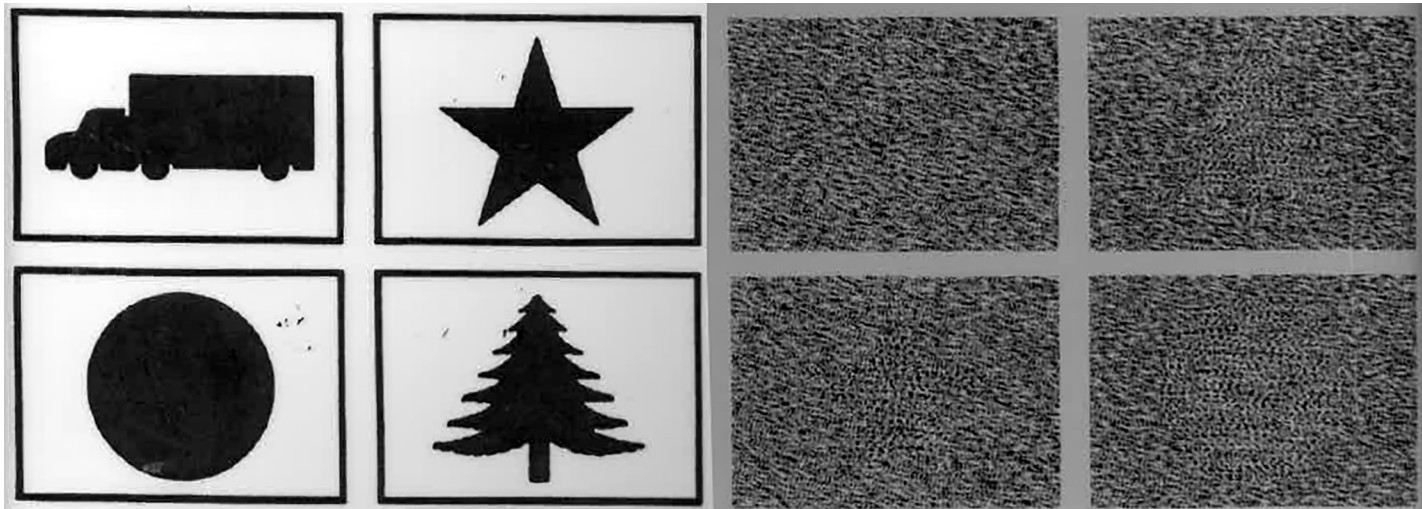

**Fig 11. Randot Preschool test card, 400 arcsec level, photographed without polarising glasses (100% crosstalk).** The disparate shapes are faintly but distinctly visible in this monocular image, e.g. one can see that (clockwise from top-left) the random dots depict blank, fir-tree, star, disk.

A further potential concern is crosstalk. The Randot stereotests use linear polarisation to separate the images for the two eyes. When used correctly, this has extremely low crosstalk. It is however critical that patients do not tilt their heads or rotate the test book, as this introduces interocular crosstalk. When crosstalk is present, both eyes can see the images intended for the separate eyes. Crosstalk generally weakens stereoscopic depth perception [40–42], but counter-intuitively it can help stereoblind observers pass the Randot Preschool test. This is because the images overlap differently in the disparate region of the image, leading to a visible difference in the combined image which can be used to pass the higher levels of the test with one eye when crosstalk is present (cf Fig 11). The crosstalk is minimal with the eyes parallel to the top edge of the book, and reaches 100% when the book is rotated through 45° relative to the interocular axis. Such crosstalk is a theoretical reason why a child without stereo vision could nevertheless achieve a measurable score on the Randot Preschoool. While we cannot rule out that this contributed to our results, we did attempt to ensure that children viewed the stereotest correctly.

**Discussion.** We find that the Randot Preschool has good specificity but poor sensitivity. This is in line with previous findings about the Randot Preschool and other stereotests. Previous results for the Randot Preschool are summarised in Table 1. Afsari et al [13] found that sensitivity was 0% when counting a "fail" as nil stereoacuity; sensitivity varied between 9% and 27% when a score of 800 arcsec also counted as a fail. Birch et al [12] found a sensitivity of 24% for the same definition of fail. Some authors have suggested that random-dot stereotests are more sensitive to strabismus than to amblyopia, but neither our data nor that of Afsari et al [13] suggests a significant difference in sensitivity (S3 and S4 Tables in S1 File). Our data and previous studies indicate that around half of children with binocular vision problems can score well with Randot Preschool, so this cannot be relied upon as a screen for binocular visual problems.

## Conclusions

We have carried out a comprehensive analysis of the Randot Preschool stereo test in a thousand children aged 2–11 years, and compared our results with previous studies. The Randot

Preschool can be successfully completed by most children from as young as 2 years old. It contains a limited range of possible stereo thresholds, which span the range of normal values in children between 2 and 5 years. These groups have a mean score of ~100 arcsec, or 2.1±0.35 $\log_{10}$ arcsec (mean±SD). In older children and adults, most people will score the best possible value, 40 arcsec, making the Randot Preschool–as the name implies–not suitable for investigating individual differences in vision in these age-groups. The Randot Preschool is extremely reliable at classifying children into those with/without any stereo vision; it is very rare for a child to pass it on one occasion and fail it one another. However, the reliability of stereo thresholds themselves is poor in the youngest age-groups, with a changes of up to ±0.7 $\log_{10}$arcsec or a factor of 5 in stereo threshold occurring by chance. Reliability is high in clinical populations, where many patients fail every time, or in older age-groups, where many obtain the best score every time. Regarded as a screen for binocular vision abnormalities such as strabismus and/or amblyopia, the Randot Preschool has excellent specificity (true negative rate >95%), meaning that almost all people without a binocular vision abnormality pass the test. Thus, failing the Randot Preschool merits further investigation. However, its sensitivity (true positive rate) is poor, <50%, so passing the Randot Preschool, even with the best possible score, certainly does not mean that a binocular vision abnormality can be ruled out.

A strength of our study is the relatively large sample and the completeness of the analysis; few previous studies have examined testability, reliability, normative values and validity. A limitation is the limited or lacking visual acuity data in the younger age-groups, and the lack of information on refraction. Using cyclopleged refraction would have enabled us to exclude anisometropic children from the normative sample, but would have required opt-in consent, which as we have seen would have likely biased the sample towards children with better vision. Different design choices have different strengths and weaknesses, which is why it is valuable to have results from many studies with different designs.

## Supporting information

**S1 File. Word document containing details of analyses referred to in the main text: (1) Normative data with more stringent limits on visual acuity; (2) Sensitivity and specificity for limited data set where full data were available; (3) Sensitivity and specificity for strabismus vs amblyopia.**
(DOCX)

## Acknowledgments

This manuscript presents independent research commissioned by the Health Innovation Challenge Fund (HICF-R8-442 and WT102565/z/13/z), a parallel funding partnership between the Wellcome Trust and the Department of Health. The views expressed in this manuscript are those of the authors and not necessarily those of the Wellcome Trust or the Department of Health.

## Disclosure

The authors have developed their own stereotest, which has recently been licensed to a company. The authors have no personal financial interest in this or any relevant product.

## Author Contributions

**Conceptualization:** Jenny C. A. Read, Kathleen Vancleef.

**Data curation:** Sheima Rafiq, Jess Hugill, Therese Casanova, Carla Black, Adam O'Neill, Vicente Puyat.

**Formal analysis:** Jenny C. A. Read, Kathleen Vancleef.

**Funding acquisition:** Jenny C. A. Read, Michael P. Clarke.

**Investigation:** Sheima Rafiq, Jess Hugill, Therese Casanova, Carla Black, Kate Taylor, Kathleen Vancleef.

**Methodology:** Jenny C. A. Read, Helen Haggerty, Kathryn Smart, Christine Powell, Kate Taylor, Michael P. Clarke, Kathleen Vancleef.

**Project administration:** Sheima Rafiq, Jess Hugill, Therese Casanova, Carla Black, Adam O'Neill.

**Resources:** Jenny C. A. Read.

**Software:** Jenny C. A. Read, Kathleen Vancleef.

**Supervision:** Kathleen Vancleef.

**Validation:** Jenny C. A. Read, Kathleen Vancleef.

**Visualization:** Jenny C. A. Read, Kathleen Vancleef.

**Writing – original draft:** Jenny C. A. Read, Kathleen Vancleef.

**Writing – review & editing:** Jenny C. A. Read, Sheima Rafiq, Jess Hugill, Therese Casanova, Carla Black, Adam O'Neill, Vicente Puyat, Helen Haggerty, Kathryn Smart, Christine Powell, Kate Taylor, Michael P. Clarke, Kathleen Vancleef.

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
