## [Decision Letter · Decision Letter 0]

29 Aug 2019

PONE-D-19-21006

Characterizing the Randot Preschool Stereotest: testability, norms, reliability, specificity and sensitivity in children aged 2-11 years

PLOS ONE

Dear Prof Read,

Thank you for submitting your manuscript to PLOS ONE. After careful consideration, we feel that it has merit but does not fully meet PLOS ONE’s publication criteria as it currently stands. Therefore, we invite you to submit a revised version of the manuscript that addresses the points raised during the review process.

Two expert reviewers have evaluated your manuscript. Reviewer 1 is very positive and provides a series of useful comments and suggestions to improve the paper. Reviewer 2 is somewhat more critical. They ask that the study be motivated better and point out a couple of potential issues regarding ceiling effects. The reviewer however also suggests how to fix these issues. I therefore expect that it should not be too difficult for the authors to address the points raised by the reviewers, and I look forward to receiving your revised work.

We would appreciate receiving your revised manuscript by Oct 13 2019 11:59PM. To enhance the reproducibility of your results, we recommend that if applicable you deposit your laboratory protocols in protocols.io, where a protocol can be assigned its own identifier (DOI) such that it can be cited independently in the future. For instructions see: http://journals.plos.org/plosone/s/submission-guidelines#loc-laboratory-protocols

We look forward to receiving your revised manuscript.

Kind regards,

Guido Maiello

Academic Editor

PLOS ONE

Journal Requirements:

The authors have developed their own stereotest, which has recently been licensed to a company. The authors have no personal financial interest in this or any relevant product.

Reviewers' comments:

Reviewer's Responses to Questions

**Comments to the Author**

1. Is the manuscript technically sound, and do the data support the conclusions?

Reviewer #1: Yes

Reviewer #2: Partly

2. Has the statistical analysis been performed appropriately and rigorously? 

Reviewer #1: Yes

Reviewer #2: I Don't Know

3. Have the authors made all data underlying the findings in their manuscript fully available?

Reviewer #1: Yes

Reviewer #2: Yes

4. Is the manuscript presented in an intelligible fashion and written in standard English?

Reviewer #1: Yes

Reviewer #2: Yes

5. Review Comments to the Author

Reviewer #1: The study evaluates the normative values, testability, and reliability of the Randot Preschool stereotest by age and sex in children 3-11. It also assesses sensitivity and specificity of the Randot Preschool to detect binocular vision issues. The strength of the study is to have a very large sample (>1000 children) and precise procedures. The manuscript is very well written, clear, and easy to read. Each result is appropriately backed up with data and statistical tests and conclusions match the data. Each result is discussed in a very interesting way. I have only a few comments, mostly minor.

Minor comments:

pp5 line 93 = « values in italics » - there is no values in italics. Also it is not clear why the ref 13 is in red, given it is not your data, isnt't it?

Pp5 – your threshold values are systematically above the other studies. I have seen that you discuss why later. I suspect it comes from the instructions: you used PEDIG instructions while the other studies may have used manufacturer instructions. The manufacturer proposes to start at 200” rather at 800”, and then to go up or down depending on success. It certainly increases the probability to be successful only by chance which lowers the average score. The PEDIG instructions certainly lead to more accurate estimations of scores.

Pp6 – line 124 : “We also excluded any reporting values solely in a clinical population, except that we did include reliability measures in clinical populations of the relevant ages.” I do not understand what you mean here.

Pp8 – line 170 and pp12 line 278 – if I understand correctly, around half of the children did not have a questionnaire filled by the time of the screening and therefore were not asked to wear glasses during testing (even if some of them might need glasses). The fact that they are not wearing glasses could explain why their stereoacuity and visual acuity is lower. Do you know what procedure other studies used regarding the correction of refraction when testing stereopsis? Could it explain the larger stereoacuity values that you obtain (if other studies actually systematically measured refraction and used frame glasses for stereopsis testing, or excluded the children with no known prescription)?

Pp13 line 293 and 300: the main result here seems to be that testability is virtually 100% after age 4. Given you have many children, would it make sense to do a statistical test (possible showing whether a confidence interval includes 1)? I think this could show that the testability is 1 at least from age x.

Pp20 – please provide a measure of goodness of fit for your model. Would you be able to justify whether it is necessary to have 8 parameters rather than 2 or 3, using any procedure to compare models (ex: AIC)? It looks to me that the distributions in Fig 8 could be described by a gamma distribution with two parameters, provided that age and sex can at least influence the scale parameter. You could compare your model to this one.

Pp23 – line 541 – missing word? “Figure” 10.

Pp23 – line 546– why include condition (c)? It does not seem to be a binocular vision problem.

Pp24 – line 588- I do not think that specificities can be considered “high” or “excellent”’ in regards to detecting participants with no binocular problem considering that the proportion of children with binocular problems is so low (4% - the target proportion). When the target proportion is that low, it is crucial that specificity is extremely high to avoid drastically decreasing the positive predictive value normalized to the target proportion. For example, if sensitivity is 100% and specificity is 96%, positive predictive value normalized to the target proportion (4%) is only 50%. Could you report positive and negative predictive values normalized to the target population as well?

Abstract: you report in the abstract that 6% of the children are stereoblind but you do not mention it in the text. Do you mean that 6% of all testable children are stereoblind or 6% of children with otherwise normal vision?

Reviewer #2: Review of Randot paper

This paper presents compressive data on the Random Preschool Stereotest administered to over 1000 children aged 2-11 years. However, its purpose is unclear because there are published data on every one of the measures the authors provide. They state their goal as getting the measures all in the same study but it is not clear why testability from one cohort and scientist and sensitivity on another cohort measured by a different scientist are any less valuable than the data presented here in separate sections of the paper for each measure (and sometimes different cohorts). The authors need to provide a stronger rationale.

The measures are also limited by a ceiling effect (no value can be tested finer than 40 arcsec). That means there is very little variance in the data from children above age 5, severely limiting the possibility of valid measures of reliability or meaningful norms. For the tests of sensitivity and specificity, the outcome will depend on the cutoffs that are used, which likely need to differ by age up to age 6—as is clear in Figure 10. This makes Figure 9 meaningless. The small number of children with binocular problems on other measures and the variety of other measures used also make the sensitivity data difficult to interpret. A brief report on the measures that are not limited by the ceiling—i.e., for the younger children only—could be of interest if the authors explain how it differs from the existing literature.

The sex difference is interesting and reminiscent of the one identified during infancy on the onset of stereopsis, although that sex difference may be an consequence of sex differences in accommodative and vergence eye movements:

Gwiazda, J., Bauer, J., & Held, R. (1989). Binocular function in human infants: correlation of stereoptic and fusion-rivalry discriminations. J Pediatr Ophthalmol Strabismus, 26(3), 128-132. Retrieved from https://www.ncbi.nlm.nih.gov/pubmed/2723974

Horwood, A. M., & Riddell, P. M. (2008). Gender differences in early accommodation and vergence development. Ophthalmic Physiol Opt, 28(2), 115-126. doi:10.1111/j.1475-1313.2008.00547.x

Minor points:

1, The authors transform all values to log units because they are more likely to be normally distributed. That’s a sound rationale but makes it very difficult to relate the findings to the actual 6 value it is possible to get on the test.

2, The acuity chart is inappropriate for children older than 4 because there are no values smaller than 0.2 loMAR. Children of age 5+ usually achieve an acuity of logMAR .10. This limits the usefulness of the analysis of sensitivity and specificity.

3, The interocular acuity difference consider indicative of abnormal binocular vision was greater than logMAR .20 (which is more than 2 lines). Amblyopia is normally defined as a difference of 2 lines or more.

4, Taking 800 arcsecs (Table 8) is an inappropriate cutoff because by age 4 almost all children score better than that.

5, Because there are only 6 discrete values and the test is not designed to determine a threshold, it’s not surprising that reliability is poor. The authors should make this point and not spend so much space analyzing the reliability data.

6. PLOS authors have the option to publish the peer review history of their article (what does this mean?). If published, this will include your full peer review and any attached files.

Reviewer #1: No

Reviewer #2: No

---

## [Author Response · Author response to Decision Letter 0]

11 Sep 2019

Response is uploaded as a separate file in order to include formatting.

---

## [Decision Letter · Decision Letter 1]

27 Sep 2019

PONE-D-19-21006R1

Characterizing the Randot Preschool Stereotest: testability, norms, reliability, specificity and sensitivity in children aged 2-11 years

PLOS ONE

Dear Prof Read,

Thank you for submitting your manuscript to PLOS ONE. After careful consideration, we feel that it has merit but does not fully meet PLOS ONE’s publication criteria as it currently stands. Therefore, we invite you to submit a revised version of the manuscript that addresses the points raised during the review process.

Reviewer 1 was satisfied with your revisions. Reviewer 2 was not available to review your revised manuscript before next month. Therefore, in order to expedite the process, I decided to assess your revisions myself. Overall, I think you did a fine job at addressing the reviewer comments, and that the manuscript has improved because of this. I’ve spotted a few remaining minor issues, which however should be very easy to address.

We would appreciate receiving your revised manuscript by Nov 11 2019 11:59PM. To enhance the reproducibility of your results, we recommend that if applicable you deposit your laboratory protocols in protocols.io, where a protocol can be assigned its own identifier (DOI) such that it can be cited independently in the future. For instructions see: http://journals.plos.org/plosone/s/submission-guidelines#loc-laboratory-protocols

We look forward to receiving your revised manuscript.

Kind regards,

Guido Maiello

Academic Editor

PLOS ONE

Additional Editor Comments (if provided):

Minor Comments:

1) I believe the x axis in figures 8 and 10 should read “Randot stereoacuity (arcsec)” instead of “Age-group”.

2) I know you have correctly uploaded your data to a public repository, but I’ve also noted that in a few places you reference analyses without reporting them. The PLOS guidelines state:

“Please note that PLOS does not permit references to “data not shown.” Authors should provide the relevant data within the manuscript, the Supporting Information files, or in a public repository. If the data are not a core part of the research study being presented, we ask that authors remove any references to these data.”

Therefore, I ask you to add these analyses as brief supporting information. Specifically, I am referring to:

Page 22 line 515: “data not shown”

Page 24 line 580: “not shown”

Page 25 Line 615 “Some authors have suggested that random-dot stereotests are more sensitive to strabismus than to amblyopia, but neither our data nor that of Afsari et al(13) suggests a significant difference in sensitivity.”

3) On page 24 line 585 you say that crosstalk “makes it possible to pass the higher levels of the test with one eye”. I think you should explain briefly how this might work, as it is not very intuitive. Additionally, you should mention that crosstalk might also worsen your results, since crosstalk decreases perceived depth:

Tsirlin, I., Wilcox, L. M., & Allison, R. S. (2012). Effect of crosstalk on depth magnitude in thin structures. Journal of Electronic Imaging, 21(1), 011003.

Reviewers' comments:

Reviewer's Responses to Questions

**Comments to the Author**

1. If the authors have adequately addressed your comments raised in a previous round of review and you feel that this manuscript is now acceptable for publication, you may indicate that here to bypass the “Comments to the Author” section, enter your conflict of interest statement in the “Confidential to Editor” section, and submit your "Accept" recommendation.

Reviewer #1: All comments have been addressed

2. Is the manuscript technically sound, and do the data support the conclusions?

Reviewer #1: Yes

3. Has the statistical analysis been performed appropriately and rigorously? 

Reviewer #1: Yes

4. Have the authors made all data underlying the findings in their manuscript fully available?

Reviewer #1: Yes

5. Is the manuscript presented in an intelligible fashion and written in standard English?

Reviewer #1: Yes

6. Review Comments to the Author

Reviewer #1: (No Response)

7. PLOS authors have the option to publish the peer review history of their article (what does this mean?). If published, this will include your full peer review and any attached files.

Reviewer #1: No

---

## [Author Response · Author response to Decision Letter 1]

8 Oct 2019

I have uploaded a separate document containing my response. In brief, I have done everything asked.

---

## [Editor Report · Decision Letter 2]

14 Oct 2019

Characterizing the Randot Preschool Stereotest: testability, norms, reliability, specificity and sensitivity in children aged 2-11 years

PONE-D-19-21006R2

Dear Dr. Read,

We are pleased to inform you that your manuscript has been judged scientifically suitable for publication and will be formally accepted for publication once it complies with all outstanding technical requirements.

With kind regards,

Guido Maiello

Academic Editor

PLOS ONE

Additional Editor Comments (optional):

A few final, extremely minor notes:

page 24 line 602: the caption of Figure 11, if the order is meant to be clockwise, I believe should read:

"the random dots depict blank, fir-tree,disk, star."

And perhaps the panels on the right side of figure 11 could be removed, they are slightly confusing as they don't match the stereograms on the right.

page 25, line 621: please reference the supplementary analyses, e.g. by amending the text to read

"Some authors have suggested that random-dot stereotests are more sensitive to strabismus than to amblyopia (Supporting Information, section 3)

Typo in the legend of table 4 of the supporting information: "amblyopiat".
---

## [Editor Report · Acceptance letter]

23 Oct 2019

PONE-D-19-21006R2 

Characterizing the Randot Preschool Stereotest: testability, norms, reliability, specificity and sensitivity in children aged 2-11 years 

Dear Dr. Read:

I am pleased to inform you that your manuscript has been deemed suitable for publication in PLOS ONE. Congratulations! Your manuscript is now with our production department. 

With kind regards,

on behalf of

Dr. Guido Maiello 

Academic Editor

PLOS ONE